# Walsh-Hadamard Variational Inference for Bayesian Deep Learning

**Simone Rossi**[*]
Data Science Department
EURECOM (FR)
simone.rossi@eurecom.fr

**Sébastien Marmin**[*]
Data Science Department
EURECOM (FR)
sebastien.marmin@eurecom.fr

**Maurizio Filippone**
Data Science Department
EURECOM (FR)
maurizio.filippone@eurecom.fr

## Abstract

Over-parameterized models, such as DeepNets and ConvNets, form a class of models that are routinely adopted in a wide variety of applications, and for which Bayesian inference is desirable but extremely challenging. Variational inference offers the tools to tackle this challenge in a scalable way and with some degree of flexibility on the approximation, but for over-parameterized models this is challenging due to the over-regularization property of the variational objective. Inspired by the literature on kernel methods, and in particular on structured approximations of distributions of random matrices, this paper proposes Walsh-Hadamard Variational Inference (WHVI), which uses Walsh-Hadamard-based factorization strategies to reduce the parameterization and accelerate computations, thus avoiding over-regularization issues with the variational objective. Extensive theoretical and empirical analyses demonstrate that WHVI yields considerable speedups and model reductions compared to other techniques to carry out approximate inference for over-parameterized models, and ultimately show how advances in kernel methods can be translated into advances in approximate Bayesian inference for Deep Learning.

## 1 Introduction

Since its inception, Variational Inference (VI, [25]) has continuously gained popularity as a scalable and flexible approximate inference scheme for a variety of models for which exact Bayesian inference is intractable. Bayesian neural networks [35, 38] represent a good example of models for which inference is intractable, and for which VI– and approximate inference in general – is challenging due to the nontrivial form of the posterior distribution and the large dimensionality of the parameter space [17, 14]. Recent advances in VI allow one to effectively deal with these issues in various ways. For instance, a flexible class of posterior approximations can be constructed using, e.g., normalizing flows [46], whereas the need to operate with large parameter spaces has pushed the research in the direction of Bayesian compression [34, 36].

Employing VI is notoriously challenging for over-parameterized statistical models. In this paper, we focus in particular on Bayesian Deep Neural Networks (DNNs) and Bayesian Convolutional Neural Networks (CNNs) as typical examples of over-parameterized models. Let's consider a supervised

---

[*]Equal contribution

learning task with $N$ input vectors and corresponding labels collected in $\boldsymbol{X} = \{\mathbf{x}_1, \ldots, \mathbf{x}_N\}$ and $\boldsymbol{Y} = \{\mathbf{y}_1, \ldots, \mathbf{y}_N\}$, respectively; furthermore, let's consider DNNs with weight matrices $\mathbf{W} = \{\boldsymbol{W}^{(1)}, \ldots, \boldsymbol{W}^{(L)}\}$, likelihood $p(\boldsymbol{Y}|\boldsymbol{X}, \mathbf{W})$, and prior $p(\mathbf{W})$. Following standard variational arguments, after introducing an approximation $q(\mathbf{W})$ to the posterior $p(\mathbf{W}|\boldsymbol{X}, \boldsymbol{Y})$ it is possible to obtain a lower bound to the log-marginal likelihood $\log[p(\boldsymbol{Y}|\boldsymbol{X})]$ as follows:

$$\log[p(\boldsymbol{Y}|\boldsymbol{X})] \geq \mathbb{E}_{q(\mathbf{W})}[\log p(\boldsymbol{Y}|\boldsymbol{X}, \mathbf{W})] - \text{KL}\{q(\mathbf{W})\|p(\mathbf{W})\}. \tag{1}$$

The first term acts as a model fitting term, whereas the second one acts as a regularizer, penalizing solutions where the posterior is far away from the prior. It is easy to verify that the KL term can be the dominant one in the objective for over-parameterized models. For example, a mean field posterior approximation turns the KL term into a sum of as many KL terms as the number of model parameters, say $Q$, which can dominate the overall objective when $Q \gg N$. As a result, the optimization focuses on keeping the approximate posterior close to the prior, disregarding the rather important model fitting term. This issue has been observed in a variety of deep models [3], where it was proposed to gradually include the KL term throughout the optimization [3, 50] to scale up the model fitting term [58, 57] or to improve the initialization of variational parameters [47]. Alternatively, other approximate inference methods for deep models with connections to VI have been proposed, notably Monte Carlo Dropout [MCD; 14] and Noisy Natural Gradients [NNG; 62].

In this paper, we propose a novel strategy to cope with model over-parameterization when using variational inference, which is inspired by the literature on kernel methods. Our proposal is to reparameterize the variational posterior over model parameters by means of a structured decomposition based on random matrix theory [54], which has inspired a number of fundamental contributions in the literature on approximations for kernel methods, such as FASTFOOD [31] and Orthogonal Random Features (ORF, [60]). The key operation within our proposal is the Walsh-Hadamard transform, and this is why we name our proposal Walsh-Hadamard Variational Inference (WHVI).

Without loss of generality, consider Bayesian DNNs with weight matrices $\boldsymbol{W}^{(l)}$ of size $D \times D$. Compared with mean field VI, WHVI has a number of attractive properties. The number of parameters is reduced from $\mathcal{O}(D^2)$ to $\mathcal{O}(D)$, thus reducing the over-regularization effect of the KL term in the variational objective. We derive expressions for the reparameterization and the local reparameterization tricks, showing that, the computational complexity is reduced from $\mathcal{O}(D^2)$ to $\mathcal{O}(D \log D)$. Finally, unlike mean field VI, WHVI induces a matrix-variate distribution to approximate the posterior over the weights, thus increasing flexibility at a log-linear cost in $D$ instead of linear.

We can think of our proposal as a specific factorization of the weight matrix, so we can speculate that other tensor factorizations [42] of the weight matrix could equally yield such benefits. Our comparison against various matrix factorization alternatives, however, shows that WHVI is superior to other parameterizations that have the same complexity. Furthermore, while matrix-variate posterior approximations have been proposed in the literature of VI [32], this comes at the expense of increasing the complexity, while our proposal keeps the complexity to log-linear in $D$.

Through a wide range of experiments on DNNs and CNNs, we demonstrate that our approach enables the possibility to run variational inference on complex over-parameterized models, while being competitive with state-of-the-art alternatives. Ultimately, our proposal shows how advances in kernel methods can be instrumental in improving VI, much like previous works showed how kernel methods can improve, e.g., Markov chain Monte Carlo sampling [48, 52] and statistical testing [18, 19, 61].

## 2 Walsh-Hadamard Variational Inference

### 2.1 Background on Structured Approximations of Kernel Matrices

WHVI is inspired by a line of works that developed from random feature expansions for kernel machines [45], which we briefly review here. A positive-definite kernel function $\kappa(\mathbf{x}_i, \mathbf{x}_j)$ induces a mapping $\phi(\mathbf{x})$, which can be infinite dimensional depending on the choice of $\kappa(\cdot, \cdot)$. Among the large literature of scalable kernel machines, random feature expansion techniques aim at constructing a finite approximation to $\phi(\cdot)$. For many kernel functions [45, 6], this approximation is built by applying a nonlinear transformation to a random projection $\boldsymbol{X}\boldsymbol{\Omega}$, where $\boldsymbol{\Omega}$ has entries $\mathcal{N}(\omega_{ij}|0, 1)$. If the matrix of training points $\boldsymbol{X}$ is $N \times D$ and we are aiming to construct $D$ random features, that is $\boldsymbol{\Omega}$ is $D \times D$, this requires $N$ times $\mathcal{O}(D^2)$ time, which can be prohibitive when $D$ is large.

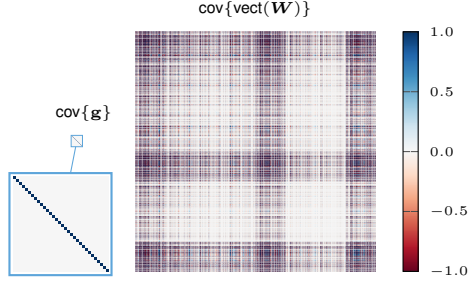

**Figure 1:** Normalized covariance of **g** and vect($\boldsymbol{W}$).

**Table 1:** Complexity of various approaches to VI

| | COMPLEXITY | |
| --- | :---: | :---: |
| | SPACE | TIME |
| MEAN FIELD GAUSSIAN | $\mathcal{O}(D^2)$ | $\mathcal{O}(D^2)$ |
| GAUSSIAN MATRIX VARIATE | $\mathcal{O}(D^2)$ | $\mathcal{O}(D^2 + M^3)$ |
| TENSOR FACTORIZATION | $\mathcal{O}(KR^2)$ | $\mathcal{O}(R^2)$ |
| WHVI | $\mathcal{O}(D)$ | $\mathcal{O}(D \log D)$ |

Note: $D$ is the dimensionality of the feature map, $K$ is the number of tensor cores, $R$ is the rank of tensor cores and $M$ is the number of pseudo-data used to sample from a matrix Gaussian distribution (see [32]).

FASTFOOD [31] tackles the issue of large dimensional problems by replacing the matrix $\boldsymbol{\Omega}$ with a random matrix for which the space complexity is reduced from $\mathcal{O}(D^2)$ to $\mathcal{O}(D)$ and time complexity of performing products with input vectors is reduced from $\mathcal{O}(D^2)$ to $\mathcal{O}(D \log D)$. In FASTFOOD, the matrix $\boldsymbol{\Omega}$ is replaced by $\boldsymbol{\Omega} \approx \boldsymbol{SHG\Pi HB}$, where $\boldsymbol{\Pi}$ is a permutation matrix, $\boldsymbol{H}$ is the Walsh-Hadamard matrix, whereas $\boldsymbol{G}$ and $\boldsymbol{B}$ are diagonal random matrices with standard Normal and Rademacher ($\{\pm 1\}$) distributions, respectively. The Walsh-Hadamard matrix is defined recursively starting from $H_2 = \begin{bmatrix} 1 & 1 \\ 1 & -1 \end{bmatrix}$ and then $H_{2D} = \begin{bmatrix} H_D & H_D \\ H_D & -H_D \end{bmatrix}$, possibly scaled by $D^{-1/2}$ to make it orthonormal. The product $\boldsymbol{Hx}$ can be computed in $\mathcal{O}(D \log D)$ time and $\mathcal{O}(1)$ space using the in-place version of the Fast Walsh-Hadamard Transform [FWHT, 12]. $\boldsymbol{S}$ is also diagonal with i.i.d. entries, and it is chosen such that the elements of $\boldsymbol{\Omega}$ obtained by this series of operations are approximately independent and follow a standard Normal (see [54] for more details). FASTFOOD inspired a series of other works on kernel approximations , whereby Gaussian random matrices are approximated by a series of products between diagonal Rademacher and Walsh-Hadamard matrices [60, 2].

## 2.2 From FASTFOOD to Walsh-Hadamard Variational Inference

FASTFOOD and its variants yield cheap approximations to Gaussian random matrices with pseudo-independent entries, and zero mean and unit variance. The question we address in this paper is whether we can use these types of approximations as cheap approximating distributions for VI. By considering a prior for the elements of the diagonal matrix $\boldsymbol{G} = \text{diag}(\mathbf{g})$ and a variational posterior $q(\mathbf{g}) = \mathcal{N}(\boldsymbol{\mu}, \boldsymbol{\Sigma})$, we can actually obtain a class of approximate posterior with some desirable properties as discussed next. Let $\boldsymbol{W} = \boldsymbol{W}^{(l)} \in \mathbb{R}^{D \times D}$ be the weight matrix of a DNN at layer $(l)$, and consider

$$\widetilde{\boldsymbol{W}} \sim q(\boldsymbol{W}) \quad \text{s.t.} \quad \widetilde{\boldsymbol{W}} = \boldsymbol{S}_1 \boldsymbol{H} \text{diag}(\tilde{\mathbf{g}}) \boldsymbol{H} \boldsymbol{S}_2 \quad \text{with} \quad \tilde{\mathbf{g}} \sim q(\mathbf{g}). \quad (2)$$

The choice of a Gaussian $q(\mathbf{g})$ and the linearity of the operations induce a parameterization of a matrix-variate Gaussian distribution for $\boldsymbol{W}$, which is controlled by $\boldsymbol{S}_1$ and $\boldsymbol{S}_2$ if we assume that we can optimize these diagonal matrices. Note that we have dropped the permutation matrix $\boldsymbol{\Pi}$ and we will show later that this is not critical for performance, while it speeds up computations.

For a generic $D_1 \times D_2$ matrix-variate Gaussian distribution, we have

$$\boldsymbol{W} \sim \mathcal{MN}(\boldsymbol{M}, \boldsymbol{U}, \boldsymbol{V}) \quad \text{if and only if} \quad \text{vect}(\boldsymbol{W}) \sim \mathcal{N}(\text{vect}(\boldsymbol{M}), \boldsymbol{V} \otimes \boldsymbol{U}), \quad (3)$$

where $\boldsymbol{M} \in \mathbb{R}^{D_1 \times D_2}$ is the mean matrix and $\boldsymbol{U} \in \mathbb{R}^{D_1 \times D_1}$ and $\boldsymbol{V} \in \mathbb{R}^{D_2 \times D_2}$ are two positive definite covariance matrices among rows and columns, and $\otimes$ denotes the Kronecker product. In WHVI, as $\boldsymbol{S}_2$ is diagonal, $\boldsymbol{HS}_2 = [\mathbf{v}_1, \dots, \mathbf{v}_D]$ with $\mathbf{v}_i = (\boldsymbol{S}_2)_{i,i}(\boldsymbol{H})_{:,i}$, so $\boldsymbol{W}$ can be rewritten in terms of $\boldsymbol{A} \in \mathbb{R}^{D^2 \times D}$ and $\mathbf{g}$ as follows

$$\text{vect}(\boldsymbol{W}) = \boldsymbol{A}\mathbf{g} \quad \text{where} \quad \boldsymbol{A}^\top = \left[ (\boldsymbol{S}_1 \boldsymbol{H} \text{diag}(\mathbf{v}_1))^\top \dots (\boldsymbol{S}_1 \boldsymbol{H} \text{diag}(\mathbf{v}_D))^\top \right]. \quad (4)$$

This rewriting, shows that the choice of $q(\mathbf{g})$ yields $q(\text{vect}(\boldsymbol{W})) = \mathcal{N}(\boldsymbol{A}\boldsymbol{\mu}, \boldsymbol{A}\boldsymbol{\Sigma}\boldsymbol{A}^\top)$, proving that WHVI assumes a matrix-variate distribution $q(\boldsymbol{W})$, see Fig. 1 for an illustration of this.

We report the expression for $M$, $U$, and $V$ and leave the full derivation to the Supplement. For the mean, we have $M = S_1 H \text{diag}(\mu) H S_2$, whereas for $U$ and $V$, we have:

$$U^{1/2} = S_1 H T_2 \quad \text{and} \quad V^{1/2} = \frac{1}{\sqrt{\text{Tr}(U)}} S_2 H T_1, \tag{5}$$

where each row $i$ of $T_1 \in \mathbb{R}^{D \times D^2}$ is the column-wise vectorization of $(\Sigma_{i,j}^{1/2} (HS_1)_{i,j'})_{j,j' \leq D}$, the matrix $T_2$ is defined similarly with $S_2$ instead of $S_1$, and $\text{Tr}(\cdot)$ denotes the trace operator.

The mean of the structured matrix-variate posterior assumed by WHVI can span a $D$-dimensional linear subspace within the whole $D^2$-dimensional parameter space, and the orientation is controlled by the matrices $S_1$ and $S_2$; more details on this geometric interpretation of WHVI can be found in the Supplement.

Matrix-variate Gaussian posteriors for variational inference have been introduced in [32]; however, assuming full covariance matrices $U$ and $V$ is memory and computationally intensive (quadratic and cubic in $D$, respectively). WHVI captures covariances across weights (see Fig. 1), while keeping memory requirements linear in $D$ and complexity log-linear in $D$.

## 2.3 Reparameterizations in WHVI for Stochastic Optimization

The so-called *reparameterization trick* [26] is a standard way to make the variational lower bound in Eq. 1 a deterministic function of the variational parameters, so as to be able to carry out gradient-based optimization despite the stochasticity of the objective. Considering input vectors $\mathbf{h}_i$ to a given layer, an improvement over this approach is to consider the distribution of the product $W \mathbf{h}_i$. This is also known as the *local reparameterization trick* [27], and it reduces the variance of stochastic gradients in the optimization, thus improving convergence. The product $W \mathbf{h}_i$ follows the distribution $\mathcal{N}(\mathbf{m}, A A^\top)$ [20], with

$$\mathbf{m} = S_1 H \text{diag}(\mu) H S_2 \mathbf{h}_i, \quad \text{and} \quad A = S_1 H \text{diag}(H S_2 \mathbf{h}_i) \Sigma^{1/2}. \tag{6}$$

A sample from this distribution can be efficiently computed thanks to the Walsh-Hadamard transform as: $\overline{W}(\mu)\mathbf{h}_i + \overline{W}(\Sigma^{1/2}\epsilon)\mathbf{h}_i$, with $\overline{W}$ a linear matrix-valued function $\overline{W}(\mathbf{u}) = S_1 H \text{diag}(\mathbf{u}) H S_2$.

## 2.4 Alternative Structures and Comparison with Tensor Factorization

The choice of the parameterization of $W$ in WHVI leaves space to several possible alternatives, which we compare in Table 2. For all of them, $G$ is learned variationally and the remaining diagonal $S_i$ (if any) are either optimized or treated variationally (Gaussian mean-field). Fig. 2 shows the behavior of these alternatives when applied to a $2 \times 64$ network with ReLU activations. With the exception of the simple and highly constrained alternative $GH$, all parameterizations are converging quite easily and the comparison with MCD shows that indeed the proposed WHVI performs better both in terms of ERROR RATE and MNLL. WHVI is effectively imposing a factorization of $W$, where parameters are either optimized or treated variationally. Tensor decompositions for DNNs and CNNs have been proposed in [42]; here $W$ is decomposed into $k$ small matrices (tensor cores), such that $W = W_1 W_2 \cdots W_k$, where each $W_i$ has dimensions $r_{i-1} \times r_i$ (with $r_1 = r_k = D$). We adapt this idea to make a comparison with WHVI. In order to match the space and time complexity of WHVI, assuming $\{r_i = R | \forall i = 2, \ldots, k-1\}$, we set: $R \propto \log_2 D$ and $K \propto \frac{D}{(\log_2 D)^2}$. Also, to

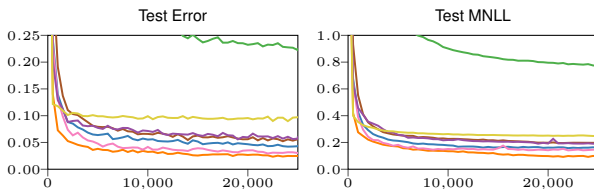

**Figure 2:** Ablation study of different structures for the parameterization of the weights distribution. Metric: test ERROR RATE and test MNLL with different structures for the weights. Benchmark on DRIVE with a $2 \times 64$ network.

**Table 2:** List of alternative structures and test performance on DRIVE dataset.

| MODEL | TEST ERROR | MNLL |
|---|---|---|
| MCD | 0.097 | 0.249 |
| $GH$ | 0.226 | 0.773 |
| $S_{\text{var}} HGH$ | 0.043 | 0.159 |
| $S_{1,\text{var}} HGH S_{2,\text{var}} H$ | 0.061 | 0.190 |
| $S_{\text{opt}} HGH$ | 0.054 | 0.199 |
| $S_{1,\text{opt}} HGH S_{2,\text{opt}} H$ | 0.031 | 0.146 |
| $S_{1,\text{opt}} HGH S_{2,\text{opt}}$ (WHVI) | **0.026** | **0.094** |

Colors are coded to match the ones used in the adjacent Figure

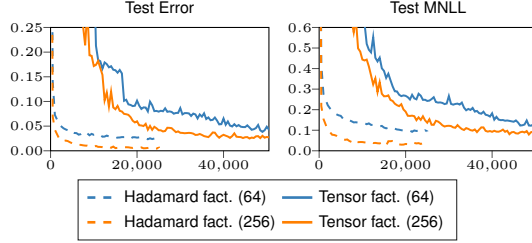

**Figure 3:** Comparison between Hadamard factorization in WHVI and tensor factorization. The number in the parenthesis is the hidden dimension. Plot is w.r.t. iterations rather then time to avoid implementation artifacts. The dataset used is DRIVE.

---

**Algorithm 1:** Setup dimensions for non-squared matrix

---
**Function** SetupDimensions($D_{\text{in}}, D_{\text{out}}$):
  next power $\leftarrow 2^{\lceil \log_2 D_{\text{in}} \rceil}$;
  **if** *next power* == $2D_{\text{in}}$ **then**
    | padding $\leftarrow 0$;
  **else**
    | padding $= $ next power $- D_{\text{in}}$;
    | $D_{\text{in}} \leftarrow$ next power;
  stack, remainder $=$ divmod($D_{\text{out}}, D_{\text{in}}$);
  **if** *remainder != 0* **then**
    | stack $\leftarrow$ stack $+ 1$;
    | $D_{\text{out}} \leftarrow D_{\text{in}} \times$ stack;
  **return** $D_{\text{in}}, D_{\text{out}}$, *padding, stack*

---

match the number of variational parameters, all internal cores ($i = 2, \ldots, k - 1$) are learned with fully factorized Gaussian posterior, while the remaining are optimized (see Table 1). Given the same asymptotic complexity, Fig. 3 reports the results of this comparison again on a 2 hidden layer network. Not only WHVI can reach better solutions in terms of test performance, but optimization is also faster. We speculate that this is attributed to the redundant variational parameterization induced by the tensor cores, which makes the optimization landscapes highly multi-modal, leading to slow convergence.

## 2.5 Extensions

**Concatenating or Reshaping Parameters for WHVI**   For the sake of presentation, so far we have assumed $\boldsymbol{W} \in \mathbb{R}^{D \times D}$ with $D = 2^d$, but we can easily extend WHVI to handle parameters of any shape $\boldsymbol{W} \in \mathbb{R}^{D_{\text{out}} \times D_{\text{in}}}$. One possibility is to use WHVI with a large $D \times D$ matrix with $D = 2^d$, such that a subset of its elements represent $\boldsymbol{W}$. Alternatively, a suitable value of $d$ can be chosen so that $\boldsymbol{W}$ is a concatenation by row/column of square matrices of size $D = 2^d$, padding if necessary (Algorithm 1 shows this case).

When one of the dimensions is equal to one so that the parameter matrix is a vector ($\boldsymbol{W} = \mathbf{w} \in \mathbb{R}^D$), this latter approach is not ideal, as WHVI would fall back on mean-field VI. WHVI can be extended to handle these cases efficiently by reshaping the parameter vector into a matrix of size $2^d$ with suitable $d$, again by padding if necessary. Thanks to the reshaping, WHVI uses $\sqrt{D}$ parameters to model a posterior over $D$, and allows for computations in $\mathcal{O}(\sqrt{D} \log D)$ rather than $D$. This is possible by reshaping the vector that multiplies the weights in a similar way. In the Supplement, we explore this idea to infer parameters of Gaussian processes linearized using large numbers of random features.

**Normalizing Flows**   Normalizing flows [NF, 46] are a family of parameterized distributions that allow for flexible approximations. In the general setting, consider a set of invertible, continuous and differentiable functions $f_k : \mathbb{R}^D \to \mathbb{R}^D$ with parameters $\boldsymbol{\lambda}_k$. Given $\mathbf{z}_0 \sim q_0(\mathbf{z}_0)$, $\mathbf{z}_0$ is transformed with a chain of $K$ flows to $\mathbf{z}_K = (f_K \circ \cdots \circ f_1)(\mathbf{z}_0)$. The variational lower bound slightly differs from Eq. 1 to take into account the determinant of the Jacobian of the transformation, yielding a new variational objective as follows:

$$\mathbb{E}_{q_0}\left[\log p(\boldsymbol{Y}|\boldsymbol{X}, \boldsymbol{W})\right] - \text{KL}\{q_0(\mathbf{z}_0)||p(\mathbf{z}_K)\} + \mathbb{E}_{q_0(\mathbf{z}_0)}\left[\sum_{k=1}^{K} \log \left|\det \frac{\partial f_k(\mathbf{z}_{k-1}; \boldsymbol{\lambda}_k)}{\partial \mathbf{z}_{k-1}}\right|\right] . \quad (7)$$

Setting the initial distribution $q_0$ to a fully factorized Gaussian $\mathcal{N}(\mathbf{z}_0|\boldsymbol{\mu}, \boldsymbol{\sigma}\mathbf{I})$ and assuming a Gaussian prior on the generated $\mathbf{z}_K$, the KL term is analytically tractable. The tranformation $f$ is generally chosen to allow for fast computation of the determinant of the Jacobian. The parameters of the initial density $q_0$ as well as the flow parameters $\boldsymbol{\lambda}$ are optimized. In our case, we consider $q_K$ as a distribution over the elements of $\mathbf{g}$. This approach increases the flexibility of the form of the variational posterior in WHVI, which is no longer Gaussian, while still capturing covariances across weights. This is obtained at the expense of losing the possibility of employing the local reparameterization trick. In the following Section, we will use *planar flows* [46]. Although this is a simple flow parameterization, a planar flow requires only $\mathcal{O}(D)$ parameters and thus it does not increase the time/space complexity of WHVI. More complex alternatives can be found in [55, 28, 33].

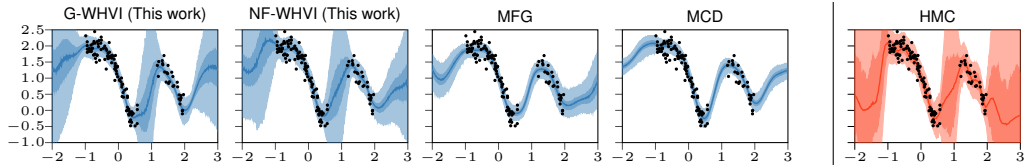

**Figure 4:** Regression example trained using WHVI with Gaussian vector (1541 param.) and with planar normalizing flow (10 flows for a total of 4141 param.), MFG (35k param.) and Monte Carlo dropout (MCD) (17k param.). The two shaded areas represent the 95th and the 75th percentile of the predictions. As "ground truth", we also show the predictive posterior obtained by running SGHMC on the same model ($R < 1.05$, [16]).

## 3   Experiments

In this Section we will provide experimental evaluations of our proposal, with experiments ranging from regression on classic benchmark datasets to image classification with large-scale convolutional neural networks. We will also comment on the computational efficiency and some potential limitation of our proposal.

### 3.1   Toy example

We begin our experimental validation with a 1D-regression problem. We generated a 1D toy regression problem with 128 inputs sampled from $\mathcal{U}[-1, 2]$, and removed 20% inputs on a predefined interval; targets are noisy realizations of a random function (noise variance $\sigma^2 = \exp(-3)$). We model these data using a DNN with 2 hidden layers of 128 features and cosine activations. We test four models: mean-field Gaussian VI (MFG), Monte Carlo dropout [MCD, 14] with dropout rate $0.4$ and two variants of WHVI – G-WHVI with Gaussian posterior and NF-WHVI with planar flows (10 planar flows). We also show the free form posterior obtained by running a MCMC algorithm, SGHMC in this case [5, 51], for several thousands steps. As Fig. 4 shows, WHVI offers a sensible modeling of the uncertainty on the input domain, whereas MFG and MCD seem to be slightly over-confident.

### 3.2   Bayesian Neural Networks

We conduct a series of comparisons with state-of-the-art VI schemes for Bayesian DNNs; see the Supplement for the list of data sets used in the experiments. We compare WHVI with MCD and NNG [NOISY-KFAC, 62]. MCD draws on a formal connection between dropout and VI with Bernoulli-like posteriors, while the more recent NOISY-KFAC yields a matrix-variate Gaussian distribution using noisy natural gradients. To these baselines, we also add the comparison with mean field Gaussian (MFG). In WHVI, the last layer assumes a fully factorized Gaussian posterior.

Data is randomly divided into 90%/10% splits for training and testing eight times. We standardize the input features **x** while keeping the targets **y** unnormalized. Differently from the experimental setup in [32, 62, 22], we use the same architecture regardless of the size of the dataset. Futhermore, to test the efficiency of WHVI in case of over-parameterized models, we set the network to have two hidden layers and 128 features with ReLU activations (as a reference, these models are ∼20 times bigger than the usual setup, which uses a single hidden layer with 50/100 units).

We report the test RMSE and the average predictive test negative log-likelihood (MNLL) in Table 3. On the majority of the datasets, WHVI outperforms MCD and NOISY-KFAC.

**Table 3:** Test RMSE and test MNLL for regression datasets. Results in the format "*mean (std)*"

| MODEL DATASET | MCD | MFG | NNG | TEST ERROR WHVI | MCD | MFG | NNG | TEST MNLL WHVI |
|---|---|---|---|---|---|---|---|---|
| BOSTON | 3.91 (0.86) | 4.47 (0.85) | 3.56 (0.43) | **3.14** (0.71) | 6.90 (2.93) | 2.99 (0.41) | **2.72** (0.09) | 4.33 (1.80) |
| CONCRETE | 5.12 (0.79) | 8.01 (0.41) | 8.21 (0.55) | **4.70** (0.72) | 3.20 (0.36) | 3.41 (0.05) | 3.56 (0.08) | **3.17** (0.37) |
| ENERGY | 2.07 (0.11) | 3.10 (0.14) | 1.96 (0.28) | **0.58** (0.07) | 4.15 (0.15) | 4.91 (0.09) | 2.11 (0.12) | **2.00** (0.60) |
| KIN8NM | 0.09 (0.00) | 0.12 (0.00) | **0.07** (0.00) | 0.08 (0.00) | −0.87 (0.02) | −0.83 (0.02) | **−1.19** (0.04) | **−1.19** (0.04) |
| NAVAL | 0.30 (0.30) | 0.01 (0.00) | **0.00** (0.00) | 0.01 (0.00) | −1.00 (2.27) | −6.23 (0.01) | **−6.52** (0.09) | −6.25 (0.01) |
| POWERPLANT | **3.97** (0.14) | 4.52 (0.13) | 4.23 (0.09) | 4.00 (0.12) | 2.74 (0.05) | 2.83 (0.03) | 2.86 (0.02) | **2.71** (0.03) |
| PROTEIN | **4.23** (0.10) | 4.93 (0.11) | 4.57 (0.47) | 4.36 (0.11) | **2.76** (0.02) | 2.92 (0.01) | 2.95 (0.12) | 2.79 (0.01) |
| YACHT | 1.90 (0.54) | 7.01 (1.22) | 5.16 (1.48) | **0.69** (0.16) | 2.95 (1.27) | 3.38 (0.29) | 3.06 (0.27) | **1.80** (1.01) |

Futhermore, we study how the test MNLL varies with the number of hidden units in a 2-layered network. As Fig. 5 shows, WHVI behaves well while competitive methods struggle. Empirically, these results demonstrate the value of WHVI, which offers a competitive parameterization of a matrix-variate Gaussian posterior while requiring log-linear time in $D$. We refer the Reader to the Supplement for additional details on the experimental setup and for the benchmark with the classic architectures.

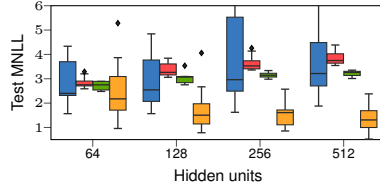

**Figure 5:** Comparison of the test MNLL as a function of the number of hidden units for MCD (●), MFG (●), NNG (●) and WHVI (●). The dataset used is YACHT.

## 3.3 Bayesian Convolutional Neural Networks

We continue the experimental evaluation of WHVI by analyzing its performance on CNNs. For this experiment, we replace all fully-connected layers in the CNN with the WHVI parameterization, while the convolutional filters are treated variationally using MCD. In this setup, we fit VGG16 [49], ALEXNET [29] and RESNET-18 [21] on CIFAR10. Using WHVI, we can reduce the number of parameters in the linear layers without affecting neither test performance nor calibration properties of the resulting model, as shown in Fig. 6 and Table 4. For ALEXNET and RESNET we also try our variant of WHVI with NF. Even though we lose the benefits of the local reparameterization, the higher flexibility of normalizing flows allows the model to obtain better test performance with respect to the Gaussian posterior. This can be improved even further using more complex families of normalizing flows [46, 55, 28, 33]. With WHVI, ALEXNET and its original ~23.3M parameters is reduced to just ~2.3M (9.9%) when using G-WHVI and to ~2.4M (10.2%) with WHVI and 3 planar flows.

**WHVI for convolutional filters** By observing that the convolution can be written as matrix multiplication (once filters are reshaped in 2D), we also extended WHVI for convolutional layers.

We observe though that in this case resulting models had too few parameters to obtain any interesting results. For ALEXNET, we obtained a model with just 189K parameters, which corresponds to a sparsity of 99.2% with respect of the original model. As a reference, Wen et al. [56] was able to reach sparsity only up to 60% in the convolutional layers without impacting performance.

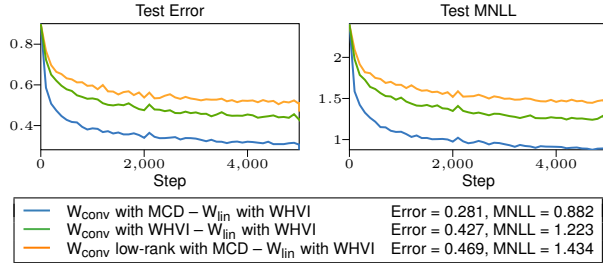

**Figure 7:** Inference of convolutional filters (dataset: CIFAR10).

To study this behavior in details, we take a simple CNN with two convolutional layers and one linear layer (Fig. 7). We see that the combination of MCD and WHVI performs very well in terms of convergence and test performance, while the use of WHVI on the convolutional filters brings an overall degradation of the performance. Interestingly, though, we also observe that MCD with the same number of parameters as for WHVI (referred to as low-rank MCD) performs even worse than the baseline: this once again confirms the parameterization of WHVI as an efficient alternative.

**Table 4:** Test performance of different Bayesian CNNs.

|  |  | TEST ERROR | TEST MNLL |
|---|---|---|---|
| VGG16 | MFG | 16.82% | 0.6443 |
|  | MCD | 21.47% | 0.8213 |
|  | NNG | 15.21% | **0.6374** |
|  | WHVI | **12.85%** | 0.6995 |
| ALEXNET | MCD | 13.30% | 0.9590 |
|  | NNG | 20.36% | – |
|  | WHVI | 13.56% | **0.6164** |
|  | NF-WHVI | **12.72%** | 0.6596 |
| RESNET18 | MCD | **10.71%** | 0.8468 |
|  | NNG | – | – |
|  | WHVI | 11.46% | 0.5513 |
|  | NF-WHVI | 11.42% | **0.4908** |

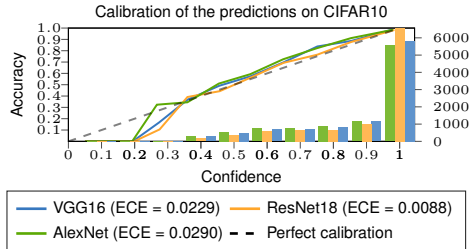

**Figure 6:** Reliability diagram and expected calibration error (ECE) of VGG16, ALEXNET and RESNET with WHVI [9, 41, 37].

### 3.4 Comments on computational efficiency

WHVI builds his computational efficiency on the Fast Walsh-Hadamard Transform (FWHT), which allows one to cut the complexity of a $D$-dimensional matrix-vector multiplication from a naive $\mathcal{O}(D^2)$ to $\mathcal{O}(D \log D)$. To empirically validate this claim, we extended PYTORCH [44] with a custom C++/CUDA kernel which implements a batched-version of the FWHT. The workstation used is equipped with two Intel Xeon CPUs, four NVIDIA Tesla P100 and 512 GB of RAM. Each experiment is carried out on a GPU fully dedicated to it. The NNG algorithm is implemented in TENSORFLOW[2] while the others are written in PYTORCH.

We made sure to fully exploit all parallelization opportunities in the competing methods and ours; we believe that the timings are not severely affected by external factors other than the actual implementation of the algorithms. The box-plots in Fig. 8 report the time required to sample and infer the carry out inference on the test set on two regression datasets as a function of the number of hidden units in a two-layer DNN. We speculate that the poor performance of NNG is due to the inversion of the approximation to the Fisher matrix, which scales cubically in the number of units.

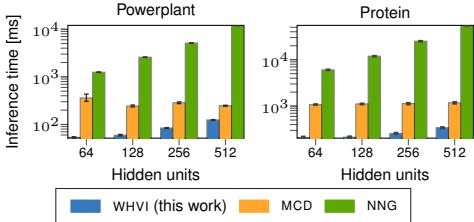

**Figure 8:** Inference time on the test set with 128 batch size and 64 Monte Carlo samples. Experiment repeated 100 times. Additional datasets available in the Supplement.

Similar behavior can also be observed for Bayesian CNNs. In Fig. 9, we analyze the energy consumption required to sample from the converged model and predict on the test set of CIFAR10 with ALEXNET using WHVI and MCD. The regularity of the algorithm for computing the FWHT and its reduced memory footprint result on an overall higher utilization of the GPU, $85\%$ for WHVI versus $\sim 70\%$ for MCD. This translates into an increase of energy efficiency up to $33\%$ w.r.t MCD, despite being $51\%$ faster.

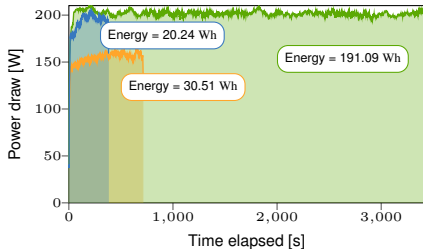

**Figure 9:** Power profiling during inference on the test set of CIFAR10 with ALEXNET and WHVI (●), MCD (●) and NNG (●). The task is repeated 16 consecutive times and profiling is carried out using the `nvidia-smi` tool.

**Additional results and insights** We refer the reader to the Supplement for an extended version of the results, including new applications of WHVI to GPs.

## Related Work

In the early sections of the paper, we have already briefly reviewed some of the literature on VI and Bayesian DNNs and CNNs; here we complement the literature by including other relevant works that have connections with WHVI.

Our work takes inspiration from the works on random features for kernel approximation [45] and FASTFOOD [31]. Random feature expansions have had a wide impact on the literature on kernel methods. Such approximations have been successfully used to scale a variety of models, such as Support Vector Machines [45], Gaussian processes [30] and Deep Gaussian processes [7, 14]. This has contributed to bridging the gap between Deep GPs and Bayesian DNNs and CNNs [38, 11, 7, 13], which is an active area of research which aims to gain a better understanding of deep learning models through the use of kernel methods [8, 10, 15]. Structured random features [31, 60, 2] have been also applied to the problem of handling large dimensional convolutional features [59] and Convolutional GPs [53].

Bayesian inference on DNNs and CNNs has been research topic of several seminar works [see e.g. 17, 22, 1, 14, 13]. Recent advances in DNNs have investigated the effect of over-parameterization and how model compression can be used during or after training [24, 34, 63]. Our current understanding shows that model performance is affected by the network size with bigger and wider neural networks

being more resilient to overfit [39, 40]. For variational inference, and Bayesian inference in general, over-parameterization is reflected on over-regularization of the objective, leading the optimization to converge to trivial solutions (posterior equal to prior). Several works have encountered and proposed solutions to such issue [23, 4, 3, 50, 47]. The problem of how to run accurate Bayesian inference on over-parametrized models like BNN is still an ongoing open question [58, 57]

## 4   Conclusions

Inspired by the literature on scalable kernel methods, this paper proposed Walsh-Hadamard Variational Inference (WHVI). WHVI offers a novel parameterization of the variational posterior, which is particularly attractive for over-parameterized models, such as modern DNNs and CNNs. WHVI assumes a matrix-variate posterior distribution, which therefore captures covariances across weights. Crucially, unlike previous work on matrix-variate posteriors for VI, this is achieved with a light parameterization and fast computations, bypassing the over-regularization issues of VI for over-parameterized models. The large experimental campaign, demonstrates that WHVI is a strong competitor of other variational approaches for such models, while offering considerable speedups.

We are currently investigating other extensions where we capture the covariance between weights across layers, by either sharing the matrix $G$ across, or by concatenating all weights into a single matrix which is then treated using WHVI, with the necessary adaptations to handle the sequential nature of computations. Finally, we are looking into deriving error bounds when using WHVI to approximate a generic matrix distribution; as preliminary work, in a numerical study in the supplement we show that the weights induced by WHVI can approximate reasonably well any arbitrary weight matrix, showing a consistent behavior w.r.t. increasing dimensions $D$.

## Broader Impact

Bayesian inference for Deep Neural Networks (DNNs) and Convolutional Neural Networks (CNNs) offers attractive solutions to many problems where one needs to combine the flexibility of these deep models with the possibility to accurately quantify uncertainty in predictions and model parameters. This is of fundamental importance in an increasingly large number of applications of machine learning in society where uncertainty matters, and where calibration of the predictions and resilience to adversarial attacks are desirable.

Due to the intractability of Bayesian inference for such models, one needs to resort to approximations. Variational inference (VI) gained popularity long before the deep learning revolution, which has seen a considerable interest in the application of VI to DNNs and CNNs in the last decade. However, VI is still under appreciated in the deep learning community because it comes with a higher computational cost for optimization, sampling, storage and inference. With this work, we offer a novel solution to this problem to make VI truly scalable in each of its parts (parameterization, sampling and inference).

Our approach is inspired by the literature on kernel methods, and we believe that this cross-fertilization will enable further contributions in both communities. In the long term, our work will make it possible to accelerate training/inference of Bayesian deep models, while reducing their storage requirements. This will complement Bayesian compression techniques to facilitate the deployment of Bayesian deep models onto FPGA, ASIC and embedded processors.

## Acknowledgments and Disclosure of Funding

The Authors would like to thanks Dino Sejdinovic for the insightful discussion on tensor decomposition, which resulted in the comparison in § 2.4. SR would like to thank Pietro Michiardi for allocating significant resources to our experimental campaign on the Zoe cloud computing platform [43]. MF gratefully acknowledges support from the AXA Research Fund and the Agence Nationale de la Recherche (grant ANR-18-CE46-0002).

## Footnotes

[1] github.com/gd-zhang/noisy-K-FAC — github.com/pomonam/NoisyNaturalGradient

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
