[Supplementary Material]

# Supplement material for "Walsh-Hadamard Variational Inference for Bayesian Deep Learning"

**Simone Rossi**[*]
Data Science Department
EURECOM (FR)
simone.rossi@eurecom.fr

**Sébastien Marmin**[*]
Data Science Department
EURECOM (FR)
sebastien.marmin@eurecom.fr

**Maurizio Filippone**
Data Science Department
EURECOM (FR)
maurizio.filippone@eurecom.fr

## A   Matrix-variate Posterior Distribution Induced by WHVI

We derive the parameters of the matrix-variate distribution $q(\boldsymbol{W}) = \mathcal{MN}(\boldsymbol{M}, \boldsymbol{U}, \boldsymbol{V})$ of the weight matrix $\tilde{\boldsymbol{W}} \in \mathbb{R}^{D \times D}$ given by WHVI,

$$\tilde{\boldsymbol{W}} = \boldsymbol{S}_1 \boldsymbol{H} \mathrm{diag}(\tilde{\mathbf{g}}) \boldsymbol{H} \boldsymbol{S}_2 \quad \text{with} \quad \tilde{\mathbf{g}} \sim \mathcal{N}(\boldsymbol{\mu}, \boldsymbol{\Sigma}). \tag{1}$$

The mean $\boldsymbol{M} = \boldsymbol{S}_1 \boldsymbol{H} \mathrm{diag}(\boldsymbol{\mu}) \boldsymbol{H} \boldsymbol{S}_2$ derives from the linearity of the expectation. The covariance matrices $\boldsymbol{U}$ and $\boldsymbol{V}$ are non-identifiable: for any scale factor $s > 0$, we have $\mathcal{MN}(\boldsymbol{M}, \boldsymbol{U}, \boldsymbol{V})$ equals $\mathcal{MN}(\boldsymbol{M}, s\boldsymbol{U}, \frac{1}{s}\boldsymbol{V})$. Therefore, we constrain the parameters such that $\mathrm{Tr}(\boldsymbol{V}) = 1$. The covariance matrices verify (see e.g. Section 1 in the supplement of [1])

$$\boldsymbol{U} = \mathbb{E}\left[(\boldsymbol{W} - \boldsymbol{M})(\boldsymbol{W} - \boldsymbol{M})^\top\right]$$
$$\boldsymbol{V} = \frac{1}{\mathrm{Tr}(\boldsymbol{U})} \mathbb{E}\left[(\boldsymbol{W} - \boldsymbol{M})^\top(\boldsymbol{W} - \boldsymbol{M})\right].$$

The Walsh-Hadamard matrix $H$ is symmetric. Denoting by $\boldsymbol{\Sigma}^{1/2}$ a root of $\boldsymbol{\Sigma}$ and considering $\boldsymbol{\epsilon} \sim \mathcal{N}(\mathbf{0}, \boldsymbol{I})$, we have

$$\boldsymbol{U} = \mathbb{E}\left[\boldsymbol{S}_1 \boldsymbol{H} \mathrm{diag}(\boldsymbol{\Sigma}^{1/2}\boldsymbol{\epsilon}) \boldsymbol{H} \boldsymbol{S}_2^2 \boldsymbol{H} \mathrm{diag}(\boldsymbol{\Sigma}^{1/2}\boldsymbol{\epsilon}) \boldsymbol{H} \boldsymbol{S}_1\right]. \tag{2}$$

If we define the matrix $\boldsymbol{T}_2 \in \mathbb{R}^{D \times D^2}$ where the $i^{\text{th}}$ row is the column-wise vectorization of the matrix $(\boldsymbol{\Sigma}_{i,j}^{1/2}(\boldsymbol{H}\boldsymbol{S}_2)_{i,j'})_{j,j' \le D}$. We have

$$(\boldsymbol{T}_2\boldsymbol{T}_2^\top)_{i,i'} = \sum_{j,j'=1}^{D} \boldsymbol{\Sigma}_{i,j}^{1/2} \boldsymbol{\Sigma}_{i',j}^{1/2} (\boldsymbol{H}\boldsymbol{S}_2)_{i,j'} (\boldsymbol{H}\boldsymbol{S}_2)_{i',j'}$$

$$= \sum_{j,j',j''=1}^{D} \boldsymbol{\Sigma}_{i,j}^{1/2} (\boldsymbol{H}\boldsymbol{S}_2)_{i,j'} \mathbb{E}[\epsilon_j \epsilon_{j''}] \boldsymbol{\Sigma}_{i',j''}^{1/2} (\boldsymbol{H}\boldsymbol{S}_2)_{i',j'}$$

$$= \sum_{j'=1}^{D} \mathbb{E}\left[\left(\sum_{j=1}^{D} \epsilon_j \boldsymbol{\Sigma}_{i,j}^{1/2} (\boldsymbol{H}\boldsymbol{S}_2)_{i,j'}\right)\left(\sum_{j''=1}^{D} \epsilon_{j''} \boldsymbol{\Sigma}_{i',j''}^{1/2} (\boldsymbol{H}\boldsymbol{S}_2)_{i',j'}\right)\right]$$

$$= \mathbb{E}\left[\left(\mathrm{diag}(\boldsymbol{\Sigma}^{1/2}\boldsymbol{\epsilon}) \boldsymbol{H} \boldsymbol{S}_2^2 \boldsymbol{H} \mathrm{diag}(\boldsymbol{\Sigma}^{1/2}\boldsymbol{\epsilon})\right)_{i,i'}\right].$$

---

[*]Equal contribution

Using (2), a root of $U = U^{1/2}U^{1/2^\top}$ can be found:

$$U^{1/2} = S_1 H T_2. \tag{3}$$

Similarly for $V$, we have

$$V^{1/2} = \frac{1}{\sqrt{\mathrm{Tr}(U)}} S_2 H T_1,$$

$$\text{with } T_1 = \begin{bmatrix} \mathrm{vect}\left(\Sigma_{1,:} (HS_1)_{1,:}^\top\right)^\top \\ \vdots \\ \mathrm{vect}\left(\Sigma_{D,:} (HS_1)_{d,:}^\top\right)^\top \end{bmatrix}. \tag{4}$$

## B   Geometric Interpretation of WHVI

The matrix $A$ in Section 2.2 expresses the linear relationship between the weights $W = S_1 HGHS_2$ and the variational random vector $\mathbf{g}$, i.e. $\mathrm{vect}(W) = A\mathbf{g}$. Recall the definition of

$$A = \begin{bmatrix} S_1 H \mathrm{diag}(\mathbf{v}_1) \\ \vdots \\ S_1 H \mathrm{diag}(\mathbf{v}_D) \end{bmatrix}, \quad \text{with } \mathbf{v}_i = (S_2)_{i,i}(H)_{:,i}. \tag{5}$$

We show that a $LQ$-decomposition of $A$ can be explicitly formulated.

**Proposition.**   Let $A$ be a $D^2 \times D$ matrix such that $\mathrm{vect}(W) = A\mathbf{g}$, where $W$ is given by $W = S_1 H \mathrm{diag}(\mathbf{g}) HS_2$. Then a $LQ$-decomposition of $A$ can be formulated as

$$\mathrm{vect}(W) = [s_i^{(2)} S_1 H \mathrm{diag}(\mathbf{h}_i)]_{i=1,\dots,D}\ \mathbf{g}$$
$$= LQ\mathbf{g}, \tag{6}$$

where $\mathbf{h}_i$ is the $i^{\text{th}}$ column of $H$, $L = \mathrm{diag}((s_i^{(2)}\mathbf{s})_{i=1,\dots,D})$, $\mathrm{diag}(\mathbf{s}^{(1)}) = S_1$, $\mathrm{diag}(\mathbf{s}^{(2)}) = S_2$, and $Q = [H\mathrm{diag}(\mathbf{h}_i)]_{i=1,\dots,D}$.

**Proof.**   *Equation (6) derives directly from block matrix and vector operations. As $L$ is clearly lower triangular (even diagonal), let us proof that $Q$ has orthogonal columns. Defining the $d \times d$ matrix $Q^{(i)} = H\mathrm{diag}(\mathbf{h}_i)$, we have:*

$$Q^\top Q = \sum_{i=1}^{D} Q^{(i)^\top} Q^{(i)}$$

$$= \sum_{i=1}^{D} \mathrm{diag}(\mathbf{h}_i) H^\top H \mathrm{diag}(\mathbf{h}_i)$$

$$= \sum_{i=1}^{D} \mathrm{diag}(\mathbf{h}_i^2) = \sum_{i=1}^{D} \frac{1}{D} I = I.$$

**Figure 1:** Diagrammatic representation of WHVI. The cube represent the high dimensional parameter space. The variational posterior (mean in orange) evolves during optimization in the (blue) subspace whose orientation (red) is controlled by $S_1$ and $S_2$.

This decomposition gives direct insight on the role of the Walsh-Hadamard transforms: with complexity $D\log(D)$, they perform fast rotations $Q$ of vectors living in a space of dimension $D$ (the plane in Fig. 1) into a space of dimension $D^2$ (the cube in Figure 1). Treated as parameters gathered in $L$, $S_1$ and $S_2$ control the orientation of the subspace by distortion of the canonical axes.

We empirically evaluate the minimum RMSE, as a proxy for some measure of average distance, between $W$ and any given point $\Gamma$. More precisely, we compute for $\Gamma \in \mathbb{R}^{D \times D}$,

$$\min_{\mathbf{s}_1, \mathbf{s}_2, \mathbf{g} \in \mathbb{R}^D} \frac{1}{D} ||\Gamma - \mathrm{diag}(\mathbf{s}_1) H \mathrm{diag}(\mathbf{g}) H \mathrm{diag}(\mathbf{s}_2)||_{\text{Frob}}. \tag{7}$$

Fig. 2 shows this quantity evaluated for $\mathbf{\Gamma}$ sampled with i.i.d $\mathcal{U}(-1, 1)$ with increasing value of $D$. The bounded behavior suggests that WHVI can approximate any given matrices with a precision that does not increase with the dimension.

**Figure 2:** Distribution of the minimum RMSE between $\mathbf{S}_1 \mathbf{H} \mathbf{G} \mathbf{H} \mathbf{S}_2$ and a sample matrix with i.i.d. $\mathcal{U}(-1, 1)$ entries. For each dimension, the orange dots represent 20 repetitions. The median distance is displayed in black. Few outliers (with distance greater than 3.0) appeared, possibly due to imperfect numerical optimization. They were kept for the calculation of the median but not displayed.

## C  Additional Details on Normalizing Flows

In the general setting, given a probabilistic model with observations $\mathbf{x}$, latent variables $\mathbf{z}$ and model parameters $\boldsymbol{\theta}$, by introducing an approximate posterior distribution $q_\phi(\mathbf{z})$ with parameters $\phi$, the variational lower bound to the log-marginal likelihood is defined as

$$
\begin{aligned}
\text{KL}\{q_\phi(\mathbf{z})||p(\mathbf{z}|\mathbf{x})\} &= \mathbb{E}_{q_\phi(\mathbf{z})}\left[\log q_\phi(\mathbf{z}) - \log p(\mathbf{z}|\mathbf{x})\right] \\
&= \mathbb{E}_{q_\phi(\mathbf{z})}\left[\log q_\phi(\mathbf{z}) - \log p_{\boldsymbol{\theta}}(\mathbf{x}, \mathbf{z}) - \log p(\mathbf{x})\right] \\
&\leq -\mathbb{E}_{q_\phi(\mathbf{z})}\left[\log p_{\boldsymbol{\theta}}(\mathbf{x}|\mathbf{z}) - \log q_\phi(\mathbf{z}) + \log p(\mathbf{z})\right]
\end{aligned} \tag{8}
$$

where $p_{\boldsymbol{\theta}}(\mathbf{x}|\mathbf{z})$ is the likelihood function with $\boldsymbol{\theta}$ model parameters and $p(\mathbf{z})$ is the prior on the latents. The objective is then to minimize the negative variational bound (NELBO):

$$
\mathcal{L}(\boldsymbol{\theta}, \phi) = -\mathbb{E}_{q_\phi(\mathbf{z})} \log p_{\boldsymbol{\theta}}(\mathbf{x}|\mathbf{z}) + \text{KL}\{q_\phi(\mathbf{z})||p(\mathbf{z}))\} \ . \tag{9}
$$

Consider an invertible, continuous and differentiable function $f : \mathbb{R}^D \to \mathbb{R}^D$. Given $\tilde{\mathbf{z}}_0 \sim q(\mathbf{z}_0)$, then $\tilde{\mathbf{z}}_1 = f(\tilde{\mathbf{z}}_0)$ follows $q(\mathbf{z}_1)$ defined as

$$
q(\mathbf{z}_1) = q(\mathbf{z}_0) \left| \det \frac{\partial f}{\partial \mathbf{z}_0} \right|^{-1} . \tag{10}
$$

As a consequence, after $K$ transformations the log-density of the final distribution is

$$
\log q(\mathbf{z}_K) = \log q(\mathbf{z}_0) - \sum_{k=1}^{K} \log \left| \det \frac{\partial f_{k-1}}{\partial \mathbf{z}_{k-1}} \right| . \tag{11}
$$

We shall define $f_k(\mathbf{z}_{k-1}; \boldsymbol{\lambda}_k)$ the $k^{\text{th}}$ transformation which takes input from the previous flow $\mathbf{z}_{k-1}$ and has parameters $\boldsymbol{\lambda}_k$. The final variational objective is

$$
\begin{aligned}
\mathcal{L}(\boldsymbol{\theta}, \phi) &= -\mathbb{E}_{q_\phi(\mathbf{z})}[\log p_{\boldsymbol{\theta}}(\mathbf{x}|\mathbf{z})] + \text{KL}\{q_\phi(\mathbf{z})||p(\mathbf{z})\}) \\
&= \mathbb{E}_{q_\phi(\mathbf{z}|\mathbf{x})}[-\log p_{\boldsymbol{\theta}}(\mathbf{x}|\mathbf{z}) - \log p(\mathbf{z}) + \log q_\phi(\mathbf{z})] \\
&= \mathbb{E}_{q_0(\mathbf{z}_0)}[-\log p_{\boldsymbol{\theta}}(\mathbf{x}|\mathbf{z}_K) - \log p(\mathbf{z}_K) + \log q_K(\mathbf{z}_K)] \\
&= \mathbb{E}_{q_0(\mathbf{z}_0)}\left[-\log p_{\boldsymbol{\theta}}(\mathbf{x}|\mathbf{z}_K) - \log p(\mathbf{z}_K) + \log q_0(\mathbf{z}_0) \right. \\
&\qquad\qquad \left. - \sum_{k=1}^{K} \log \left| \det \frac{\partial f_k(\mathbf{z}_{k-1}; \boldsymbol{\lambda}_k)}{\partial \mathbf{z}_{k-1}} \right| \right] \\
&= -\mathbb{E}_{q_0(\mathbf{z}_0)} \log p_{\boldsymbol{\theta}}(\mathbf{x}|\mathbf{z}) + \text{KL}\{q_0(\mathbf{z}_0)||p(\mathbf{z}_K)\}
\end{aligned}
$$

$$-\mathbb{E}_{q_0(\mathbf{z}_0)} \sum_{k=1}^{K} \log \left| \det \frac{\partial f_k(\mathbf{z}_{k-1}; \boldsymbol{\lambda}_k)}{\partial \mathbf{z}_{k-1}} \right| . \tag{12}$$

Setting the initial distribution $q_0$ to a fully factorized Gaussian $\mathcal{N}(\mathbf{z}_0|\boldsymbol{\mu}, \boldsymbol{\sigma}\mathbf{I})$ and assuming a Gaussian prior on the generated $\mathbf{z}_K$, the KL term is analytically tractable. A possible family of transformation is the *planar flow* [10]. For the *planar flow*, $f$ is defined as

$$f(\mathbf{z}) = \mathbf{z} + \mathbf{u}h(\mathbf{w}^\top \mathbf{z} + b), \tag{13}$$

where $\lambda = [\mathbf{u} \in \mathbb{R}^D, \mathbf{w} \in \mathbb{R}^D, b \in \mathbb{R}]$ and $h(\cdot) = \tanh(\cdot)$. This is equivalent to a residual layer with single neuron MLP – as argued by Kingma et al. [5]. The log-determinant of the Jacobian of $f$ is

$$\log \left| \det \frac{\partial f}{\partial \mathbf{z}} \right| = \left| \det(\mathbf{I} + \mathbf{u}[h'(\mathbf{w}^\top \mathbf{z} + b)\mathbf{w}]^\top) \right|$$
$$= \left| 1 + \mathbf{u}^\top \mathbf{w} h'(\mathbf{w}^\top \mathbf{z} + b) \right| . \tag{14}$$

Although this is a simple flow parameterization, a planar flow requires only $\mathcal{O}(D)$ parameters and thus it does not increase the time/space complexity of WHVI. Alternatives can be found in [10, 13, 5, 8].

## D   Additional Results

### D.1   Experimental Setup for Bayesian DNN

The experiments on Bayesian DNN are run with the following setup. For WHVI, we used a zero-mean prior over $\mathbf{g}$ with fully factorized covariance $\lambda\boldsymbol{I}$; $\lambda = 10^{-5}$ was chosen to obtain sensible variances in the output layer. It is possible to design a prior over $\mathbf{g}$ such that the prior on $\boldsymbol{W}$ has constant marginal variance and low correlations although empirical evaluations showed not to yield a significant improvement compared to the previous (simpler) choice. In the final implementation of WHVI that we used in all experiments, $\boldsymbol{S}_1$ and $\boldsymbol{S}_2$ are optimized. The dropout rate of MCD is set to 0.005. We used classic Gaussian likelihood with optimized noise variance for regression and softmax likelihood for classification.

**Table 1:** List of dataset used in the experiments

| NAME | TASK | N. | D-IN | D-OUT |
|---|---|---|---|---|
| EEG | CLASS. | 14980 | 14 | 2 |
| MAGIC | CLASS. | 19020 | 10 | 2 |
| MINIBOO | CLASS. | 130064 | 50 | 2 |
| LETTER | CLASS. | 20000 | 16 | 26 |
| DRIVE | CLASS. | 58509 | 48 | 11 |
| MOCAP | CLASS. | 78095 | 37 | 5 |
| CIFAR10 | CLASS. | 60000 | $3 \times 28 \times 28$ | 10 |
| BOSTON | REGR. | 506 | 13 | 1 |
| CONCRETE | REGR. | 1030 | 8 | 1 |
| ENERGY | REGR. | 768 | 8 | 2 |
| KIN8NM | REGR. | 8192 | 8 | 1 |
| NAVAL | REGR. | 11934 | 16 | 2 |
| POWERPLANT | REGR. | 9568 | 4 | 1 |
| PROTEIN | REGR. | 45730 | 9 | 1 |
| YACHT | REGR. | 308 | 6 | 1 |
| BOREHOL | REGR. | 200000 | 8 | 1 |
| HARTMAN6 | REGR. | 30000 | 6 | 1 |
| RASTRIGIN5 | REGR. | 10000 | 5 | 1 |
| ROBOT | REGR. | 150000 | 8 | 1 |
| OTLCIRCUIT | REGR. | 20000 | 6 | 1 |

Training is performed for 500 steps with fixed noise variance and for other 50000 steps with optimized noise variance. Batch size is fixed to 64 and for the estimation of the expected loglikelihood we used 1 Monte Carlo sample at train-time and 64 Monte Carlo samples at test-time. We choose the Adam optimizer [4] with exponential learning rate decay $\lambda_{t+1} = \lambda_0(1 + \gamma t)^{-p}$, with $\lambda_0 = 0.001$, $p = 0.3$, $\gamma = 0.0005$ and $t$ being the current iteration.

Similar setup was also used for the Bayesian CNN experiment. The only differences are the batch size – increased to 256 – and the optimizer, which is run without learning rate decay.

## D.2 Regression Experiments on Shallow Models

For a complete experimental evaluation of WHVI, we also use the experimental setup proposed by Hernandez-Lobato and Adams [3] and adopted in several other works [2, 7, 14]. In this configuration, we use one hidden layer with 50 hidden units for all datasets with the exception of PROTEIN where the number of units is increased to 100. Results are reported in Table 2.

**Table 2:** Test RMSE and test MNLL for regression datasets following the setup in [3].

| MODEL DATASET | TEST ERROR | | | | TEST MNLL | | | |
|---|---|---|---|---|---|---|---|---|
| | MCD | MFG | NNG | WHVI | MCD | MFG | NNG | WHVI |
| BOSTON | 3.40 (0.66) | 3.04 (0.64) | 2.74 (0.12) | 2.56 (0.15) | 5.04 (1.76) | 3.19 (0.89) | 2.45 (0.03) | 2.55 (0.15) |
| CONCRETE | 4.60 (0.53) | 5.24 (0.53) | 5.02 (0.12) | 5.01 (0.25) | 2.96 (0.23) | 3.03 (0.15) | 3.04 (0.02) | 2.95 (0.06) |
| ENERGY | 1.18 (0.03) | 1.52 (0.09) | 0.48 (0.02) | 1.20 (0.07) | 3.00 (0.07) | 3.49 (0.11) | 1.42 (0.00) | 3.01 (0.12) |
| KIN8NM | 0.09 (0.00) | 0.10 (0.00) | 0.08 (0.00) | 0.12 (0.01) | $-1.09$ (0.04) | $-1.01$ (0.04) | $-1.15$ (0.00) | $-0.78$ (0.10) |
| NAVAL | 0.00 (0.00) | 0.01 (0.00) | 0.00 (0.00) | 0.01 (0.00) | $-9.93$ (0.01) | $-6.48$ (0.02) | $-7.08$ (0.03) | $-6.25$ (0.01) |
| POWERPLANT | 4.20 (0.12) | 4.23 (0.13) | 3.89 (0.04) | 4.11 (0.12) | 2.76 (0.03) | 2.77 (0.03) | 2.78 (0.01) | 2.74 (0.03) |
| PROTEIN | 4.35 (0.04) | 4.74 (0.05) | 4.10 (0.00) | 4.64 (0.07) | 2.80 (0.01) | 2.89 (0.01) | 2.84 (0.00) | 2.86 (0.01) |
| YACHT | 1.72 (0.32) | 1.78 (0.45) | 0.98 (0.08) | 0.96 (0.20) | 2.73 (0.74) | 2.02 (0.46) | 2.32 (0.00) | 1.28 (0.22) |

## D.3 ConvNets architectures

**Figure 3:** Architecture layout of RESNET 18.

For the experiments on Bayesian convolutional neural networks, we used architectures adapted to CIFAR10 (see Tables 3, 4 and 5).

**Table 3:** ALEXNET

| LAYER | DIMENSIONS |
|---|---|
| CONV | $64 \times 3 \times 3 \times 3$ |
| MAXPOOL | |
| CONV | $192 \times 64 \times 3 \times 3$ |
| MAXPOOL | |
| CONV | $384 \times 192 \times 3 \times 3$ |
| CONV | $256 \times 384 \times 3 \times 3$ |
| CONV | $256 \times 256 \times 3 \times 3$ |
| MAXPOOL | |
| LINEAR | $4096 \times 4096$ |
| LINEAR | $4096 \times 4096$ |
| LINEAR | $10 \times 4096$ |

**Table 4:** VGG16

| LAYER | DIMENSIONS |
|---|---|
| CONV | $32 \times 3 \times 3 \times 3$ |
| CONV | $32 \times 32 \times 3 \times 3$ |
| MAXPOOL | |
| CONV | $64 \times 32 \times 3 \times 3$ |
| CONV | $64 \times 64 \times 3 \times 3$ |
| MAXPOOL | |
| CONV | $128 \times 64 \times 3 \times 3$ |
| CONV | $128 \times 128 \times 3 \times 3$ |
| CONV | $128 \times 128 \times 3 \times 3$ |
| MAXPOOL | |
| CONV | $256 \times 128 \times 3 \times 3$ |
| CONV | $256 \times 256 \times 3 \times 3$ |
| CONV | $256 \times 256 \times 3 \times 3$ |
| MAXPOOL | |
| CONV | $256 \times 256 \times 3 \times 3$ |
| CONV | $256 \times 256 \times 3 \times 3$ |
| CONV | $256 \times 256 \times 3 \times 3$ |
| MAXPOOL | |
| LINEAR | $10 \times 256$ |

**Table 5:** RESNET 18

| LAYER | DIMENSIONS |
|---|---|
| RESNET BLOCK | $\begin{bmatrix} 3 \times 3, 64 \\ 3 \times 3, 64 \end{bmatrix} \times 2$ |
| RESNET BLOCK | $\begin{bmatrix} 3 \times 3, 128 \\ 3 \times 3, 128 \end{bmatrix} \times 2$ |
| RESNET BLOCK | $\begin{bmatrix} 3 \times 3, 256 \\ 3 \times 3, 256 \end{bmatrix} \times 2$ |
| RESNET BLOCK | $\begin{bmatrix} 3 \times 3, 512 \\ 3 \times 3, 512 \end{bmatrix} \times 2$ |
| AVGPOOL | |
| LINEAR | $10 \times 512$ |

**Table 6:** Complexity table for GPs with random feature and inducing points approximations. In the case of random features, we include both the complexity of computing random features and the complexity of treating the linear combination of the weights variationally (using VI and WHVI).

| | COMPLEXITY | |
| | SPACE | TIME |
| --- | --- | --- |
| MEAN FIELD - RF | $\mathcal{O}(D_{\text{IN}}N_{\text{RF}}) + \mathcal{O}(N_{\text{RF}}D_{\text{OUT}})$ | $\mathcal{O}(D_{\text{IN}}N_{\text{RF}}) + \mathcal{O}(N_{\text{RF}}D_{\text{OUT}})$ |
| WHVI - RF | $\mathcal{O}(D_{\text{IN}}N_{\text{RF}}) + \mathcal{O}(\sqrt{N_{\text{RF}}}D_{\text{OUT}})$ | $\mathcal{O}(D_{\text{IN}}N_{\text{RF}}) + \mathcal{O}(D_{\text{OUT}}\log N_{\text{RF}})$ |
| INDUCING POINTS | $\mathcal{O}(M)$ | $\mathcal{O}(M^3)$ |

Note: $M$ is the number of pseudo-data/inducing points and $N_{RF}$ is the number of random features used in the kernel approximation.

## D.4 Results - Gaussian Processes with Random Feature Expansion

We test WHVI for scalable GP inference, by focusing on GPs with random feature expansions [6]. In GP models, latent variables $\mathbf{f}$ are given a prior $p(\mathbf{f}) = \mathcal{N}(\mathbf{0}|\boldsymbol{K})$; the assumption of zero mean can be easily relaxed. Given a random feature expansion of the kernel martix, say $\boldsymbol{K} \approx \boldsymbol{\Phi}\boldsymbol{\Phi}^\top$, the latent variables can be rewritten as:

$$\mathbf{f} = \boldsymbol{\Phi}\mathbf{w} \tag{15}$$

with $\mathbf{w} \sim \mathcal{N}(\mathbf{0}, \boldsymbol{I})$. The random features $\boldsymbol{\Phi}$ are constructed by randomly projecting the input matrix $\boldsymbol{X}$ using a Gaussian random matrix $\boldsymbol{\Omega}$ and applying a nonlinear transformation, which depends on the choice of the kernel function. The resulting model is now linear, and considering regression problems such that $\mathbf{y} = \mathbf{f} + \boldsymbol{\varepsilon}$ with $\boldsymbol{\varepsilon} \sim \mathcal{N}(\mathbf{0}, \sigma^2\boldsymbol{I})$, solving GPs for regression becomes equivalent to solving standard linear regression problems. For a given set of random features, we treat the weights of the resulting linear layer variationally and evaluate the performance of WHVI.

By reshaping the vector of parameters $\mathbf{w}$ of the linear model into a $D \times D$ matrix, WHVI allows for the linearized GP model to reduce the number of parameters to optimize (see Table 6). We compare WHVI with two alternatives; one is VI of the Fourier features GP expansion that uses less random features to match the number of parameters used in WHVI, and another is the sparse Gaussian process implementation of GPFLOW [9] with a number of inducing points (rounded up) to match the number of parameters used in WHVI.

We report the results on five datasets ($10000 \leq N \leq 200000$, $5 \leq D \leq 8$, see Table 1). The data sets are generated from space-filling evaluations of well known functions in analysis of computer

**Figure 4:** Comparison of test error w.r.t. the number model parameters (*top:* mean field, *bottom:* full covariance).

experiments (see e.g. [12]). Dataset splitting in training and testing points is random uniform, 20% versus 80 %. The input variables are rescaled between 0 and 1. The output values are standardized for training. All GPs have the same prior (centered GP with RBF covariance), initialized with equal hyperparameter values: each of the $D$ lengthscale to $\sqrt{D/2}$, the GP variance to 1, the Gaussian likelihood standard deviation to 0.02 (prior observation noise). The training is performed with 12000 steps of Adam optimizer. The observation noise is fixed for the first 10000 steps. Learning rate is $6 \times 10^{-4}$, except for the dataset HARTMAN6 with a learning rate of $5 \times 10^{-3}$. Sparse GPs are run with whitened representation of the inducing points.

The results are shown in Fig. 4 with diagonal covariance for the three variational posteriors and with full covariance. In both mean field and full covariance settings, this variant of WHVI using the reshaping of $W$ into a column largely outperforms the direct VI of Fourier features. However, it appears that this improvement of the random feature inference for GPs is not enough to reach the performance of VI using inducing points. Inducing point approximations are based on the Nyströom approximation of kernel matrices, which are known to lead to lower approximation error on the elements on the kernel matrix compared to random features approximations. This is the reason we attribute to the lower performance of WHVI compared to inducing points approximations in this experiment.

### D.5   Extended results - DNNs

Being able to increase width and depth of a model without drastically increasing the number of variational parameters is one of the competitive advantages of WHVI. Fig. 5 shows the behavior of WHVI for different network configurations. At test time, increasing the number of hidden layers and the numbers of hidden features allow the model to avoid overfitting while delivering better performance. This evidence is also supported by the analysis of the test MNLL during optimization of the ELBO, as showed in Fig. 6.

Thanks to WHVI structure of the weights matrices, expanding and deepening the model is beneficial not only at convergence but during the entire learning procedure as well. Furthermore, the derived NELBO is still a valid lower bound of the true marginal likelihood and, therefore, a suitable objective function for model selection. Differently from the issue addressed in [11], during our experiments we didn't experience problems regarding initialization of variational parameters. We claim that this is possible thanks to both the reduced number of parameters and the effect of the Walsh-Hadamard transform.

**Timing profiling of the Fast Walsh-Hadamard transform**   Key to the log-linear time complexity is the Fast Walsh-Hadamard transform, which allows to perform the operation $Hx$ in $\mathcal{O}(D \log D)$ time without requiring to generate and store $H$. For our experimental evaluation, we implemented a FWHT operation in PYTORCH (v. 0.4.1) in C++ and CUDA to leverage the full computational capabilities of modern GPUs. Fig. 8 presents a timing profiling of our implementation versus the naive matmul (batch size of 512 samples and profiling repeated 1000 times). The breakeven point for the CPU implementation is in the neighborhood of 512/1024 features, while on GPU we see FWHT is consistently faster.

**Figure 5:** Analysis of model capacity for different features and hidden layers.

**Figure 6:** Comparison of test performance. Being able to increase features and hidden layers without worrying about overfitting/overparametrize the model is advantageous not only at convergence but during the entire learning procedure

**Figure 7:** Inference time on the test set with 128 batch size and 64 Monte Carlo samples. Experiment repeated 100 times. Additional datasets available in the Supplement.

**Figure 8:** On the (**left**), time performance versus number of features (D) with batch size fixed to 512. On the (**right**) distribution of inference time versus batch size (D =512) with MATMUL and FWHT on GPU.

**Table 7:** Test error of Bayesian DNN with 2 hidden layers on regression datasets. NF: number of hidden features

| MODEL | DATASET NF | BOSTON | CONCRETE | ENERGY | KIN8NM | NAVAL | POWERPLANT | PROTEIN | TEST ERROR YACHT |
|---|---|---|---|---|---|---|---|---|---|
| MCD | 64 | $3.80 \pm 0.88$ | $5.43 \pm 0.69$ | $2.13 \pm 0.12$ | $0.17 \pm 0.22$ | $0.07 \pm 0.00$ | — | $4.36 \pm 0.12$ | $2.02 \pm 0.51$ |
| | 128 | $3.91 \pm 0.86$ | $5.12 \pm 0.79$ | $2.07 \pm 0.11$ | $0.09 \pm 0.00$ | $0.30 \pm 0.30$ | $3.97 \pm 0.14$ | $4.23 \pm 0.10$ | $1.90 \pm 0.54$ |
| | 256 | $3.62 \pm 1.01$ | $5.03 \pm 0.74$ | $2.04 \pm 0.11$ | $0.10 \pm 0.00$ | $0.07 \pm 0.00$ | $3.91 \pm 0.11$ | $4.09 \pm 0.11$ | $2.09 \pm 0.66$ |
| | 512 | $3.56 \pm 0.85$ | $4.81 \pm 0.79$ | $2.03 \pm 0.12$ | $0.09 \pm 0.00$ | $0.07 \pm 0.00$ | $\mathbf{3.90 \pm 0.10}$ | $\mathbf{3.87 \pm 0.11}$ | $2.09 \pm 0.55$ |
| MFG | 64 | $4.06 \pm 0.72$ | $6.87 \pm 0.54$ | $2.42 \pm 0.12$ | $0.11 \pm 0.00$ | $0.01 \pm 0.00$ | $4.38 \pm 0.12$ | $4.85 \pm 0.12$ | $4.31 \pm 0.62$ |
| | 128 | $4.47 \pm 0.85$ | $8.01 \pm 0.41$ | $3.10 \pm 0.14$ | $0.12 \pm 0.00$ | $0.01 \pm 0.00$ | $4.52 \pm 0.13$ | $4.93 \pm 0.11$ | $7.01 \pm 1.22$ |
| | 256 | $5.27 \pm 0.98$ | $9.41 \pm 0.54$ | $4.03 \pm 0.10$ | $0.13 \pm 0.00$ | $0.01 \pm 0.00$ | $4.79 \pm 0.12$ | $5.07 \pm 0.12$ | $8.71 \pm 1.31$ |
| | 512 | $6.04 \pm 0.90$ | $10.84 \pm 0.46$ | $4.90 \pm 0.11$ | $0.16 \pm 0.00$ | $0.01 \pm 0.00$ | $5.53 \pm 0.16$ | $5.26 \pm 0.10$ | $10.34 \pm 1.45$ |
| NNG | 64 | $3.20 \pm 0.26$ | $6.90 \pm 0.59$ | $1.54 \pm 0.18$ | $0.07 \pm 0.00$ | $\mathbf{0.00 \pm 0.00}$ | $3.94 \pm 0.05$ | $3.90 \pm 0.02$ | $3.57 \pm 0.70$ |
| | 128 | $3.56 \pm 0.43$ | $8.21 \pm 0.55$ | $1.96 \pm 0.28$ | $0.07 \pm 0.00$ | $\mathbf{0.00 \pm 0.00}$ | $4.23 \pm 0.09$ | $4.57 \pm 0.47$ | $5.16 \pm 1.48$ |
| | 256 | $4.87 \pm 0.94$ | $8.18 \pm 0.57$ | $3.41 \pm 0.55$ | $0.07 \pm 0.00$ | $\mathbf{0.00 \pm 0.00}$ | $4.07 \pm 0.00$ | $4.88 \pm 0.00$ | $5.60 \pm 0.65$ |
| | 512 | $5.19 \pm 0.62$ | $11.67 \pm 2.06$ | $5.12 \pm 0.37$ | $0.10 \pm 0.00$ | $\mathbf{0.00 \pm 0.00}$ | $4.97 \pm 0.00$ | — | $5.91 \pm 0.80$ |
| WHVI | 64 | $3.33 \pm 0.82$ | $5.24 \pm 0.77$ | $0.73 \pm 0.11$ | $0.08 \pm 0.00$ | $0.01 \pm 0.00$ | $4.07 \pm 0.11$ | $4.49 \pm 0.12$ | $0.82 \pm 0.18$ |
| | 128 | $3.14 \pm 0.71$ | $4.70 \pm 0.72$ | $0.58 \pm 0.07$ | $0.08 \pm 0.00$ | $0.01 \pm 0.00$ | $4.00 \pm 0.12$ | $4.36 \pm 0.11$ | $0.69 \pm 0.16$ |
| | 256 | $2.99 \pm 0.85$ | $4.63 \pm 0.78$ | $0.52 \pm 0.07$ | $0.08 \pm 0.00$ | $0.01 \pm 0.00$ | $3.95 \pm 0.12$ | $4.24 \pm 0.11$ | $0.76 \pm 0.13$ |
| | 512 | $\mathbf{2.99 \pm 0.69}$ | $\mathbf{4.51 \pm 0.80}$ | $\mathbf{0.51 \pm 0.04}$ | $\mathbf{0.07 \pm 0.00}$ | $0.01 \pm 0.00$ | $3.96 \pm 0.12$ | $4.14 \pm 0.09$ | $\mathbf{0.71 \pm 0.16}$ |

**Table 8:** Test MNLL of Bayesian DNN with 2 hidden layers on regression datasets. NF: number of hidden features

| MODEL | DATASET NF | BOSTON | CONCRETE | ENERGY | KIN8NM | NAVAL | POWERPLANT | PROTEIN | TEST MNLL YACHT |
|---|---|---|---|---|---|---|---|---|---|
| MCD | 64 | $5.67 \pm 2.35$ | $3.19 \pm 0.28$ | $4.19 \pm 0.15$ | $-0.78 \pm 0.69$ | $-2.68 \pm 0.00$ | — | $2.79 \pm 0.01$ | $2.85 \pm 1.02$ |
| | 128 | $6.90 \pm 2.93$ | $3.20 \pm 0.36$ | $4.15 \pm 0.15$ | $-0.87 \pm 0.02$ | $-1.00 \pm 2.27$ | $2.74 \pm 0.05$ | $2.76 \pm 0.02$ | $2.95 \pm 1.27$ |
| | 256 | $6.60 \pm 3.59$ | $3.31 \pm 0.45$ | $4.13 \pm 0.15$ | $-0.70 \pm 0.05$ | $-2.70 \pm 0.00$ | $2.75 \pm 0.04$ | $2.72 \pm 0.01$ | $3.79 \pm 1.88$ |
| | 512 | $7.28 \pm 3.31$ | $3.45 \pm 0.59$ | $4.13 \pm 0.17$ | $-0.76 \pm 0.03$ | $-2.71 \pm 0.00$ | $2.77 \pm 0.04$ | $\mathbf{2.68 \pm 0.02}$ | $3.76 \pm 1.65$ |
| MFG | 64 | $2.83 \pm 0.33$ | $3.26 \pm 0.08$ | $4.42 \pm 0.10$ | $-0.92 \pm 0.02$ | $-6.24 \pm 0.01$ | $2.80 \pm 0.03$ | $2.90 \pm 0.01$ | $2.85 \pm 0.24$ |
| | 128 | $2.99 \pm 0.41$ | $3.41 \pm 0.05$ | $4.91 \pm 0.09$ | $-0.83 \pm 0.02$ | $-6.23 \pm 0.01$ | $2.83 \pm 0.01$ | $2.92 \pm 0.01$ | $3.38 \pm 0.29$ |
| | 256 | $3.33 \pm 0.53$ | $3.57 \pm 0.07$ | $5.44 \pm 0.05$ | $-0.69 \pm 0.01$ | $-6.22 \pm 0.01$ | $2.89 \pm 0.02$ | $2.95 \pm 0.01$ | $3.65 \pm 0.32$ |
| | 512 | $3.69 \pm 0.54$ | $3.73 \pm 0.05$ | $5.83 \pm 0.05$ | $-0.49 \pm 0.01$ | $-6.19 \pm 0.01$ | $3.04 \pm 0.03$ | $2.98 \pm 0.01$ | $3.86 \pm 0.31$ |
| NNG | 64 | $\mathbf{2.69 \pm 0.06}$ | $3.40 \pm 0.15$ | $\mathbf{1.95 \pm 0.08}$ | $-1.14 \pm 0.05$ | $-5.83 \pm 1.49$ | $2.80 \pm 0.01$ | $2.78 \pm 0.01$ | $2.71 \pm 0.17$ |
| | 128 | $2.72 \pm 0.09$ | $3.56 \pm 0.08$ | $2.11 \pm 0.12$ | $-1.19 \pm 0.04$ | $\mathbf{-6.52 \pm 0.09}$ | $2.86 \pm 0.02$ | $2.95 \pm 0.12$ | $3.06 \pm 0.27$ |
| | 256 | $3.04 \pm 0.22$ | $3.52 \pm 0.07$ | $2.64 \pm 0.17$ | $-1.19 \pm 0.03$ | $-5.73 \pm 0.21$ | $2.84 \pm 0.01$ | $3.02 \pm 0.01$ | $3.15 \pm 0.13$ |
| | 512 | $3.13 \pm 0.14$ | $3.91 \pm 0.20$ | $3.07 \pm 0.07$ | $-0.80 \pm 0.00$ | $-5.30 \pm 0.05$ | $3.51 \pm 0.00$ | — | $3.21 \pm 0.14$ |
| WHVI | 64 | $3.68 \pm 1.40$ | $3.19 \pm 0.34$ | $2.18 \pm 0.37$ | $-1.13 \pm 0.02$ | $-6.25 \pm 0.01$ | $2.73 \pm 0.03$ | $2.82 \pm 0.01$ | $2.56 \pm 1.33$ |
| | 128 | $4.33 \pm 1.80$ | $\mathbf{3.17 \pm 0.37}$ | $2.00 \pm 0.60$ | $-1.19 \pm 0.04$ | $-6.25 \pm 0.01$ | $2.71 \pm 0.03$ | $2.79 \pm 0.01$ | $1.80 \pm 1.01$ |
| | 256 | $4.99 \pm 2.65$ | $3.35 \pm 0.59$ | $2.06 \pm 0.72$ | $\mathbf{-1.23 \pm 0.04}$ | $-6.25 \pm 0.01$ | $\mathbf{2.70 \pm 0.03}$ | $2.77 \pm 0.01$ | $1.53 \pm 0.53$ |
| | 512 | $5.41 \pm 2.30$ | $3.33 \pm 0.56$ | $2.05 \pm 0.46$ | $-1.22 \pm 0.04$ | $-6.25 \pm 0.01$ | $\mathbf{2.70 \pm 0.03}$ | $2.74 \pm 0.01$ | $\mathbf{1.37 \pm 0.57}$ |

**Table 9:** Results of Bayesian DNN on 6 classification datasets. Note: NL: number of hidden layers, NF: number of hidden features

| MODEL | NL | DATASET NF | TEST ERROR | | | | | | TEST MNLL | | | | | |
|---|---|---|---|---|---|---|---|---|---|---|---|---|---|---|
| | | | DRIVE | EEG | LETTER | MAGIC | MINIBOO | MOCAP | DRIVE | EEG | LETTER | MAGIC | MINIBOO | MOCAP |
| MCD | 2 | 64 | 0.19 ± 0.11 | 0.16 ± 0.01 | 0.45 ± 0.05 | 0.13 ± 0.02 | **0.07 ± 0.00** | 0.02 ± 0.02 | 0.52 ± 0.24 | 0.36 ± 0.02 | 1.27 ± 0.26 | 0.37 ± 0.12 | 0.18 ± 0.00 | 0.11 ± 0.10 |
| | | 128 | 0.17 ± 0.07 | 0.19 ± 0.11 | 0.45 ± 0.04 | 0.16 ± 0.08 | 0.15 ± 0.21 | 0.04 ± 0.07 | 0.47 ± 0.19 | 0.36 ± 0.09 | 1.39 ± 0.22 | 0.33 ± 0.04 | 0.24 ± 0.17 | 0.10 ± 0.11 |
| | | 256 | 0.16 ± 0.09 | 0.20 ± 0.15 | 0.45 ± 0.06 | **0.13 ± 0.01** | 0.07 ± 0.00 | 0.16 ± 0.13 | 0.50 ± 0.29 | **0.33 ± 0.08** | 1.32 ± 0.25 | 0.35 ± 0.09 | **0.17 ± 0.00** | 0.29 ± 0.21 |
| | | 512 | 0.18 ± 0.11 | 0.18 ± 0.15 | 0.44 ± 0.02 | 0.18 ± 0.10 | **0.07 ± 0.00** | 0.03 ± 0.06 | 0.47 ± 0.27 | 0.95 ± 1.63 | 1.41 ± 0.17 | 0.40 ± 0.06 | 0.20 ± 0.04 | 0.17 ± 0.22 |
| | 3 | 64 | 0.34 ± 0.10 | **0.13 ± 0.01** | 0.50 ± 0.06 | 0.16 ± 0.07 | 0.08 ± 0.02 | 0.09 ± 0.09 | 0.88 ± 0.25 | 0.55 ± 0.61 | 1.56 ± 0.28 | 0.42 ± 0.16 | 0.20 ± 0.05 | 0.18 ± 0.15 |
| | | 128 | 0.32 ± 0.10 | 0.21 ± 0.14 | 0.48 ± 0.09 | 0.16 ± 0.07 | 0.23 ± 0.28 | 0.11 ± 0.19 | 0.86 ± 0.28 | 1.46 ± 2.78 | 1.40 ± 0.34 | 0.44 ± 0.13 | 0.28 ± 0.18 | 0.34 ± 0.28 |
| | | 256 | 0.32 ± 0.21 | 0.23 ± 0.17 | 0.43 ± 0.05 | 0.14 ± 0.00 | 0.23 ± 0.28 | 0.28 ± 0.26 | 0.87 ± 0.51 | 0.40 ± 0.09 | 1.34 ± 0.19 | 0.62 ± 0.07 | 0.31 ± 0.22 | 0.61 ± 0.48 |
| | | 512 | 0.36 ± 0.09 | 0.14 ± 0.11 | 0.49 ± 0.06 | 0.14 ± 0.01 | 0.23 ± 0.28 | 0.23 ± 0.12 | 0.93 ± 0.27 | 0.74 ± 0.78 | 1.92 ± 0.23 | 1.02 ± 0.15 | 0.30 ± 0.20 | 0.45 ± 0.27 |
| WHVI | 2 | 64 | 0.03 ± 0.01 | 0.25 ± 0.01 | 0.43 ± 0.01 | **0.13 ± 0.01** | 0.10 ± 0.00 | 0.08 ± 0.01 | 0.14 ± 0.04 | 0.61 ± 0.28 | 1.07 ± 0.02 | 0.32 ± 0.02 | 0.23 ± 0.01 | 0.28 ± 0.02 |
| | | 128 | 0.02 ± 0.00 | 0.21 ± 0.01 | 0.41 ± 0.01 | **0.13 ± 0.01** | 0.09 ± 0.00 | 0.05 ± 0.00 | 0.09 ± 0.02 | 0.45 ± 0.01 | 1.02 ± 0.02 | 0.32 ± 0.02 | 0.22 ± 0.01 | 0.17 ± 0.01 |
| | | 256 | **0.01 ± 0.00** | 0.19 ± 0.01 | 0.40 ± 0.01 | **0.13 ± 0.01** | 0.08 ± 0.00 | 0.03 ± 0.00 | 0.09 ± 0.03 | 0.76 ± 0.92 | 0.99 ± 0.01 | 0.31 ± 0.02 | 0.20 ± 0.00 | 0.12 ± 0.01 |
| | | 512 | **0.01 ± 0.00** | 0.17 ± 0.01 | 0.40 ± 0.01 | **0.13 ± 0.01** | 0.08 ± 0.00 | **0.02 ± 0.00** | 0.08 ± 0.03 | 0.52 ± 0.37 | 0.97 ± 0.01 | **0.31 ± 0.01** | 0.19 ± 0.01 | 0.08 ± 0.01 |
| | 3 | 64 | 0.03 ± 0.00 | 0.33 ± 0.05 | 0.42 ± 0.01 | **0.13 ± 0.01** | 0.10 ± 0.00 | 0.07 ± 0.01 | 0.12 ± 0.02 | 0.61 ± 0.05 | 1.02 ± 0.02 | 0.32 ± 0.01 | 0.23 ± 0.01 | 0.24 ± 0.02 |
| | | 128 | 0.02 ± 0.00 | 0.38 ± 0.09 | 0.41 ± 0.01 | **0.13 ± 0.01** | 0.09 ± 0.00 | 0.04 ± 0.00 | 0.09 ± 0.02 | 0.64 ± 0.07 | 0.98 ± 0.01 | 0.31 ± 0.02 | 0.22 ± 0.01 | 0.15 ± 0.01 |
| | | 256 | 0.05 ± 0.09 | 0.45 ± 0.01 | 0.39 ± 0.01 | **0.13 ± 0.01** | 0.08 ± 0.00 | **0.02 ± 0.00** | 0.20 ± 0.34 | 0.69 ± 0.00 | 0.94 ± 0.02 | 0.31 ± 0.02 | 0.20 ± 0.01 | 0.09 ± 0.01 |
| | | 512 | **0.01 ± 0.00** | 0.45 ± 0.01 | **0.38 ± 0.01** | **0.13 ± 0.01** | 0.08 ± 0.00 | **0.02 ± 0.00** | **0.05 ± 0.02** | 0.69 ± 0.00 | **0.90 ± 0.01** | 0.32 ± 0.01 | 0.19 ± 0.01 | **0.06 ± 0.01** |