[Reviews · NeurIPS 2020]

Review 1

Summary and Contributions: This paper proposes a novel method for variational inference in deep neural networks which uses a Walsh-Hadamard-based factorization, effcectively resulting in a matrix-variate Gaussian posterior approximation, to reduce the number of parameters and speed up inference. This structured posterior parameterization also avoids the well-known issue of over-regularization of (mean-field) VI in over-parameterized models. The paper also shows that the Walsh-Hadamard factorization of the weight matrices is superior to other factorizations with the same complexity, and describes extensions of the approach to leverage normalizing flows for more expressive inference. An empirical evaluation on several benchmark demonstrates the efficacy of the proposed WHVI approach, as compared to previous methods for approximate Bayesian inference in deep neural networks.

Strengths: This paper tackles an important problem in machine learning, namely estimating distributions over the weights of deep neural network models, which has many critical real-world applications. The central novelty of this paper is the idea of using a Walsh-Hadamard factorization to parameterize a matrix-variate Gaussian variational posterior approximation, which is favorable in terms of memory and time complexity as compared to alternative approaches. The proposed idea is principled yet simple, and seems to be both efficient and effective on various benchmark problems. The paper is certainly relevant to the NeurIPS community, and presents significant contributions that will be of interest to researchers and practitioners both working on deep learning and probabilistic modeling.

Weaknesses: The main weakness I see with this paper is its empirical evaluation, which could be more convincing. While the experiments on CNNs show that WHVI is competitive with other approaches on VGG16 while being more parameter efficient (which is impressive), I am not sure how well this is aligned with the goal of the paper. I was under the impression that the goal of the paper was to improve Bayesian inference in deep neural networks (for which I would expect stronger results), but instead the goal might be to reduce the number of model parameters without sacrificing accuracy -- it would be great if the authors could clarify this. Furthermore, I would have liked to see a more extensive evaluation of uncertainty calibration, both in in-domain and especially out-of-domain settings, using e.g. the benchmarks proposed in Ovadia et al. 2019, which would further strenghten the paper. Also, the paper does not compare against state-of-the-art methods for deep uncertainty quantification such as deep ensembles (Lakshminarayanan et al. 2017, Ovadia et al. 2019), which makes it hard to assess the potential impact of the proposed approach. Another interesting baseline would have been the non-Bayesian versions of the neural networks, i.e. fit using MAP estimation, in order to see the benefit of the Bayesian approach. Another weakness is that the approach, in its current form, is only applicable to fully connected layers, not to e.g. convolutional filters, which is a clear limitation given the prevalence and importance of convolutional architectures in deep learning. In the experiments, the authors use a combination of WHVI over the fully connected layers and Monte Carlo Dropout over the convolutional filters for convolutional architectures, which seems a bit hacky and less principled. References: Ovadia et al., "Can you trust your model's uncertainty? Evaluating predictive uncertainty under dataset shift", NeurIPS 2019 Lakshminarayanan et al., "Simple and scalable predictive uncertainty estimation using deep ensembles", NeurIPS 2017

Correctness: Yes, the claims, derivations and empirical methodology appears to be correct, as far as I can tell.

Clarity: Yes, the paper is overall very well written and easy to follow.

Relation to Prior Work: Yes, the paper discusses related prior work and contrasts it to the proposed approach to an adequate degree.

Reproducibility: No

Additional Feedback: POST-REBUTTAL Thank you for commenting to some of the concerns and questions I had raised. After a detailed read of the other reviews and the authors response, my overall opinion of the paper has not changed. I still agree with the other reviewers that the proposed approach is interesting and promising, but that the empirical evaluation could be significantly strengthened. Furthermore, the limitation that the proposed approach is only really applicable to fully-connected layers was not commented on in the author response, which makes me inclined to believe that the authors are trying not to draw too much attention to this, potentially indicating that they agree that this is a significant limitation. At the very least, I would like to see a more thorough and honest discussion of this limitation. Modern architectures for vision tasks are predominately composed of convolutional layers and often only have very few fully-connected layers (as also pointed out by reviewer #2). It might well be the case that doing inference only over the last few fully-connected layers is enough, but again, this should be discussed and/or assessed in a bit more detail. For example, it would be very interesting if the author would compare their approach to the so-called "neural linear" approach which performs inference over only the last fully-connected layer of a neural network [1,2]. While I am still a bit more inclined to see the paper accepted, I am not convinced enough to be willing to fight for acceptance due to the issues raised. [1] Riquelme et al., 2018, "Deep Bayesian Bandits Showdown: An Empirical Comparison of Bayesian Deep Networks for Thompson Sampling" [2] Ober et al., 2019, "Benchmarking the Neural Linear Model for Regression" ============= Questions: - Section 3.1: why did you use SGHMC instead of full HMC, which should be tractable on this small-scale example? - Section 3.2: why do you assme a fully factorized Gaussian posterior for the last layer in WHVI? - Section 3.2: do you have any intuition as to why WHVI outperforms the other methods on certain datasets, but not on others? in which settings should one expect WHVI to work well, and in which to not work well? Minor issues: - Table 1 is not referenced in the text - Figure 1: it would be useful if the caption contained the main take-away of this figure, which is not immediately clear without consulting the text - l. 141: the acronym MNLL is not defined - the Related Work section should probably be numbered


Review 2

Summary and Contributions: This paper proposes a low-rank variational distribution for the parameters of BNNs using the Walsh-Hadamard factorization. They derive efficient inference techniques and show that the method works on some benchmark data sets. Update: After reading the author response, I still think the experiments section could be significantly strengthened, but I won't fight against accepting the paper.

Strengths: - VI for BNNs is still a hard problem and innovations like this one are needed to convince practitioners to switch from standard NNs to Bayesian ones. - The Walsh-Hadamard factorization is theoretically well motivated and seems to have a favorable tradeoff between flexibility and parameter count.

Weaknesses: - The CNN experiments are not fully convincing (see below). - Some related work is not properly addressed (see below).

Correctness: The claims and method seem to be correct.

Clarity: The paper is well written.

Relation to Prior Work: It seems like some related works have been missed (which might be parallel work, but could still be mentioned). Especially [1] and [2] also use a low-rank posterior approximation and could be useful baselines to compare to or at least discuss in relation to the proposed method. In more general terms, [3] discusses inference on subspaces which is also related since the Walsh-Hadamard factorization implicitly induces a subspace on the weight space. It could be fruitful to discuss the geometry of this induced subspace w.r.t. the ones proposed in [3]. [1] Swiatkowski, J., Roth, K., Veeling, B. S., Tran, L., Dillon, J. V., Mandt, S., ... & Nowozin, S. (2020). The k-tied Normal Distribution: A Compact Parameterization of Gaussian Mean Field Posteriors in Bayesian Neural Networks. arXiv preprint arXiv:2002.02655. [2] Dusenberry, M. W., Jerfel, G., Wen, Y., Ma, Y. A., Snoek, J., Heller, K., ... & Tran, D. (2020). Efficient and Scalable Bayesian Neural Nets with Rank-1 Factors. arXiv preprint arXiv:2005.07186. [3] Izmailov, P., Maddox, W. J., Kirichenko, P., Garipov, T., Vetrov, D., & Wilson, A. G. (2019). Subspace inference for Bayesian deep learning. arXiv preprint arXiv:1907.07504.

Reproducibility: Yes

Additional Feedback: - In the CNN experiment, how many layers of the different models were actually treated with WHVI? Because it says that it was only applied to the fully connected ones and many CNN architectures only have a single fully connected layer... - In Fig. 5, the variance in performance seems to be larger for WHVI than for NNG and MFG. Why is that? - Fig. 6 only shows the calibration for WHVI, right? How does it compare to the other methods?


Review 3

Summary and Contributions: The authors proposed a novel parameterization method for variational inference, inspired by the scalable kernel methods as random feature expansions and FASTFOOD (Sarlos et al., 2013). The core idea is using the Walsh-Hadamard transform for the weight matrix of Bayesian DNN; therefore, the authors named the proposed method as Walsh-Hadamard Variational Inference (WHVI). They showed that WHVI could reduce the number of parameters from $O(D^{2})$ to $O(D})$, which means that this method could deal with the over-regularization problem for KL term caused from the deep Bayesian neural network in the objective function of variational inference. They also showed that WHVI could reduce the computational complexity from $O(D^{2})$ to $O(D\logD)$, by deriving the expression for the reparameterization and the local reparameterization trick. Furthermore, they showed WHVI could be compatible with the other flexible approximation schemes, such as normalizing flows. Finally, they confirmed the performance of WHVI through three experiments by comparing the benchmark methods and revealed that WHVI outperformed the mean-field Gaussian VI (MFG), Monte Carlo dropout (MCD), and Noisy Natural Gradient (NNG) in terms of uncertainty evaluation (compared with MFG and MCD), predictive performances, and computation efficiency.

Strengths: ・Although the idea itself is a simple and straightforward application of FASTFOOD and therefore not so novel, there are some great properties in the proposed method (can significantly reduce {the dimensions of weights; the computational complexity}). ・It seems that WHVI is easy to implement and can compatible with flexible approximation methods such as normalizing flows. ・The authors confirmed the performances and usefulness of WHVI through a large number of experiments (for uncertainty evaluation, predictive performance, and computational efficiency). WHVI seems to outperform the benchmark methods on several datasets.

Weaknesses: Although this paper is well-written, and the proposed method seems useful for Bayesian deep learning, I had some concerns as follows. (a) The impact of the idea I think the proposed method is just derived from a straightforward application of the concept of kernel approximation scheme (FASTFOOD). Of course, the simple and effective methodology is ideal in the real world, but I concern that, in the sense of the idea used in, the contribution in this paper is less comparable to the level the NeurIPS community wants. Is not there any other better kernel approximation methods can be applied for? Why is FASTFOOD the best choice for parameter reduction and computational complexity reduction? The concrete answer/explanation for these questions may strengthen the claim of this paper. (b) Experiments I think there are enough experiments for predictive performances and computational efficiency. However, it seems that there are no answers/explanations why the proposed method can offer sensible modeling of the uncertainty on the input domain, and the other methods can not do so. As the authors said in the broader impact section, uncertainty quantification is a critical issue in society, and bayesian DNN has been paid attention to in this point of view. Therefore, it makes strong this paper to explain the mechanism of the proposed method in the uncertainty quantification perspective and to conduct more experiments (in appendix due to the paper limitation) at this point. Because of these concerns, I decided to set the overall score of this paper as 6~7.

Correctness: It seems that the claims and the empirical methodology are reasonable. In this paper, there are many insightful assertions based on the detailed examples and the reasonable experiments from the many points of view.

Clarity: The structure and organization of the presentation are good. The explanation of contribution is clear and easy to follow the related work. The contents in the appendix are really helpful to understand (especially the geometric interpretation is interesting).

Relation to Prior Work: The related work section is well-written. The contribution of this paper obviously differs from the previous contributions in terms of the computational efficiency and the coping method for over-parameterization & over-regularization in Bayesian deep learning. However, the idea and technique itself have already well investigated on kernel methods.

Reproducibility: Yes

Additional Feedback: All of the comments, suggestions, and questions are in the above sections. ========================= After reading the author response ========================= I read all of the reviews and the authors' rebuttal. My concerns have almost been allayed, although it is necessary to improve the description of the proposed method's contributions and novelty in this paper. Therefore, I decided to increase my score. However, as the other reviewers said, the experimental part in this paper could be more strengthened. Furthermore, if the author(s) make the statement for the contribution of this paper and the uncertainty evaluation clearer in the final version like that of the authors' feedback, this paper will be a more good paper to read.


Review 4

Summary and Contributions: The paper introduces a Walsh-Hadamard factorized form for the approximate posterior of the weights of Bayesian neural network. This has the advantage of reducing the effect of regularization due to overparameterization and at the same time reducing computational complexity. Edit: Upon careful consideration, I decided to increase my score to 7 as the limitations of this paper do not outweight its strenghts.

Strengths: - While the use of tensor factorization in the variational approximation of Bayesian neural nets posterior is not new, the use of the very scalable Walsh-Hadamard form is particularly suitable for the large scale application that are common in ML and deep learning. In particular, the almost linear time complexity is crucial in wide network applications where the quadratic scaling would become intractable. - It is interesting to see cross fertilization between the bayesian neural nets and the kernel literature. These connections have deep root as in the wide net limit Bayesian NNs often "converge" to Gaussian processes. The present work can be an useful stepping stone to further connect these fields both algorithmically and theoretically. - The experiment section of the paper cover a quite large range of applications including regression and classification in both low dimensional and computer vision problems. The result are encouraging and seems to suggest that the proposed method has higher performance than the most relevant baselines at least for small size networks and low-D problems.

Weaknesses: - The experiment section is below acceptance threshold in my opinion. This paper offers a new very scalable technique that it is very appropriate for large scale experiments with wide networks. However, most of the experiments are very low-dimensional and they do not really probe the over-parameterized regime that should be target of the paper. The largest scale experiment uses decently small networks on CIFAR10 and it does not provide much insight into the regime where the proposed method would be the most useful. The conv net experiment are poorly described and inconclusive and their treatment should be improved. - The main idea to reduce over-regularization by factorizing the variational distribution is somewhat artificial from a Bayesian point of view as it would be more principled to act directly on the prior. However, from a practical standpoint the proposed approach makes sense as far as it is corroborated by comprehensive experimental evidence.

Correctness: The main empirical limitation of the paper is that its experiments focus on relatively simple low-D problems while the rationale of the method is targeted at large networks where both the reduction of over-regularization and the favorable scaling becomes crucial. Without these large scale experiments, it is very difficult to evaluate the performance of the method against other factorized baselines with similar scaling properties.

Clarity: The paper is well written with a clear explanation of the rationale of the method and of the relevant background theory. the experiments section is a little chaotic and its structure could be improved. The section describing the ConvNet experiment is quite incomplete and it should be either improved or omitted.

Relation to Prior Work: The prior work on factorized variational approximation and feature expansion is clearly discussed.

Reproducibility: Yes

Additional Feedback: A larger scale experiment on a dataset such as ImageNet would help but it is not required. What is mostly missing is a killer application on a currently important problem. It is clear that the method gives benefits for fully connected networks but those architectures are hardly used nowadays.

[Author Response · NeurIPS 2020]

First of all we would like to thank the Reviewers for their valuable comments and useful suggestions, which will be taken into account for the revised version. With this paper we propose a novel method (Walsh-Hadamard Variational Inference) that enables VI to scale on models for which inference is known to be challenging (Bayesian DNNs and CNNs). Extensions to non-Gaussian approximate posterior (using normalizing flows) and to Gaussian processes are also highlighted in the main paper as well as in the Supplement. We are delighted to see that all the Reviewers acknowledge the novelty and the potential impact of our contribution for the NeurIPS community, as well as the overall clarity of the paper. Below, we address the main points raised by the Reviewers.

**Reviewer 1:** Thank you for acknowledging the significance of our contribution and its relevance for researches and pratictioners. *"I'm not sure how well this is aligned with the goal of the paper"*. The goal is actually twofold: we show that we can reduce the parameterization of such models, and we do it in such a way that performance not only don't degradate, but they are even enhanced. For VGG16 for example, this means going from 15% of error rate with NNG down to 12% with WHVI. *"I would have liked [...] evaluation of uncertainty calibration"*. Thanks for the suggestion and the reference. Looking at the Ovadia et al. (2019) implementation of SVI (mean-field Gaussian), we expect to behave similarly or better. *"The paper does not compare with [...] deep ensembles [...] non-Bayesian versions of the neural networks"*. We believe deep ensembles are a bit out of the scope of the comparison but we are happy to include them in the revised paper. *"Why did you use SGHMC instead of full HMC?"*. It was for convenience. But for this simple example we don't expect it to make big difference (the R-hat statistic showed its convergence). We will add some traces and the setup in the supplement. *"Why do you assume a fully factorized Gaussian posterior [...]?"* Our parameterization in case of output dimension 1 will be equivalent to mean field. WHVI can be extended to handle these cases by reshaping the parameter vector into a matrix (see experiment with GPs). *"In which settings should one expect WHVI to work well?"* We expect WHVI to work progressively better the deeper and the wider a model is. From the point of view of the parameterization, in the main paper and in the supplement we discuss possible limitations of the proposal and we show a numerical study where we evaluate the average distance of any random matrix to the closest WHVI matrix, showing constant behavior with increasing dimensions. *"Table 1 is not referenced in the text"*. Thanks, there should be a reference in §2.4 *"Caption of Figure 1 [...]. The acronym MNLL [...]. Related Work section [...]"*. Thanks for the suggestion, we will fix that.

**Reviewer 2:** *"Some related work have been missed"*. Thanks for pointing out these references, which are actually very related to our work. We will include them in the final revision. *"Setup of the CNN experiment"*. We'll point the reviewer to the supplement §D for a detailed explanation of the experimental setup. *"The variance seems larger for WHVI [...]"*. In our experimental results, WHVI does not show systematically larger variance than other methods (see the full tabular version). For this case, we argue this is due to both having a very small dataset (the smallest, in fact) and different initializations. *"Fig. 6 shows the calibration for WHVI?"* Correct.

**Reviewer 3:** We are thrilled that you enjoyed reading our work! *"Is not there any other better kernel approximation methods?"* In Fig. 2 and Table 2, we have experimented with different parameterizations which can be related to different kernel approximation [58, 2]. We will make this statement more clear in the final version. *"Why is FASTFOOD the best choice [...]?"* As we wrote in the introduction, FASTFOOD was mostly a starting point to develop our method. Nonetheless, we tried different structures to confirm that indeed what we propose is a sensible parameterization. We believe that the superior performance comes from the sequence of projections due to the Hadamard matrix $\mathbf{H}$. Following our geometrical argument (see supplement), the Walsh-Hadamard transforms perform fast rotations of vectors living in a space of dimension $D$ in a space $D \times D$ with complexity $D \log D$. *"It seems that there are no answers/explanations why the proposed method can offer sensible modeling of the uncertainty"*. The cause is actually twofold: richer variational approximation and efficient parameterization. We know that from the point of view of achieving good approximations to the true posterior distribution, the mean-field family (like fully factorized Gaussian) is possibly one of the roughest approximations we can do. This is commonly done due to their simple implementation and (relative) good predictive performance. On the contrary, the posterior induced by WHVI on $\mathbf{W}$ is a matrix-variate Gaussian (in the vanilla case) with non-diagonal covariances: this is definitely more expressive than the fully factorized case.

**Reviewer 4:** *"The main idea [...] is somewhat artificial from a Bayesian point of view as it would be more principled to act directly on the prior."* Nice point! We agree that from a Bayesian p.o.v., the prior is generally the way to go. On the other hand, for these kind of models acting directly on the prior (of the parameters) is somehow very complicated and still an open problem. Recent proposals like downplaying the KL [22, 3, 48] or using temperature scaling do in fact act on the prior (implicitly, like we do) but in artificial ways (decreasing the regularization strength of the KL or increasing the magnitude of the likelihood). What we propose is developed on the solid ground of kernel methods.

As a way to characterize this behavior, we analyze the shape of the functional prior on $\mathbf{f}$. On the right, a simple visualization for a 4-layer network with 64 hidden units, where we show the pathologies of a simple Gaussian prior versus our parameterization. Thanks again for this interesting comment. We will expand on this in the final version.

*"The experiments section is a little chaotic"*. Thanks for the suggestion, we will re-structure this section a bit, thus making it easier to follow.

[Meta-Review · NeurIPS 2020]

Walsh-Hadamard factorizations for variational posteriors are proposed. While reviewers appreciated the paper, the discussion brought to light several concerns shared across the reviewers. R1 in particular has updated their review to reflect some of these points from the discussion. It seems the proposed approaches are only applicable to a fully-connected last layer. There was a sense in the discussion that the authors had dodged these questions rather than addressing them directly. Last layer methods are certainly useful, and widely used in practice, such that this (significant) constraint would certainly be acceptable, if directly and honestly presented, alongside comparisons to such methods, such as the references [1,2] provided by R1. The reviewers did like the basic ideas and thought they could help open new directions of inquiry. The authors are strongly advised to carefully consider the reviews, and their updates, in revising the manuscript.